# Exact and Soft Successive Refinement of the Information Bottleneck

**DOI:** 10.3390/e25091355

**Published:** 2023-09-19

**Authors:** Hippolyte Charvin, Nicola Catenacci Volpi, Daniel Polani

**Affiliations:** School of Physics, Engineering and Computer Science, University of Hertfordshire, Hatfield AL10 9AB, UK; n.catenacci-volpi@herts.ac.uk (N.C.V.); d.polani@herts.ac.uk (D.P.)

**Keywords:** information bottleneck, successive refinement, unique information, incremental learning, coarse-graining, Blackwell order, deep learning

## Abstract

The information bottleneck (IB) framework formalises the essential requirement for efficient information processing systems to achieve an optimal balance between the complexity of their representation and the amount of information extracted about relevant features. However, since the representation complexity affordable by real-world systems may vary in time, the processing cost of updating the representations should also be taken into account. A crucial question is thus the extent to which adaptive systems can *leverage the information content of already existing IB-optimal representations for producing new ones*, which target the same relevant features but at a different granularity. We investigate the information-theoretic optimal limits of this process by studying and extending, within the IB framework, the notion of *successive refinement*, which describes the ideal situation where no information needs to be discarded for adapting an IB-optimal representation’s granularity. Thanks in particular to a new geometric characterisation, we analytically derive the successive refinability of some specific IB problems (for binary variables, for jointly Gaussian variables, and for the relevancy variable being a deterministic function of the source variable), and provide a linear-programming-based tool to numerically investigate, in the discrete case, the successive refinement of the IB. We then soften this notion into a *quantification* of the loss of information optimality induced by several-stage processing through an existing measure of unique information. Simple numerical experiments suggest that this quantity is typically low, though not entirely negligible. These results could have important implications for (i) the structure and efficiency of incremental learning in biological and artificial agents, (ii) the comparison of IB-optimal observation channels in statistical decision problems, and (iii) the IB theory of deep neural networks.

## 1. Introduction

### 1.1. Conceptualisation and Organisation Outline

Consider the problem, for an information-processing system, of extracting relevant information about a target variable *Y* within a correlated source variable *X*, under constraints on the cost of the information processing needed to do so—yielding a compressed representation *T*. This situation can be formalised in an information-theoretic language, where the information-processing cost is measured with the mutual information I(X;T) between the source *X* and the representation *T* of it, while the relevancy about *Y* of the information extracted by *T* is measured by I(Y;T). The problem thus becomes that of maximising the relevant information I(Y;T) under bounded information-processing cost I(X;T), i.e., we are interested in the *information bottleneck* (IB) problem [1,2], which, in primal form, can be formulated as
(1)$$\begin{array}{c} \underset{\begin{array}{c} q(T|X)\;:\\ T-X-Y,\;I(X;T)\leq \lambda \end{array}}{arg max}\;I\left(Y;T\right). \end{array}$$ Here, the trade-off parameter λ controls the bound on the permitted information-processing cost and thus, intuitively, the resulting representation’s granularity. The Markov chain condition T−X−Y ensures that any information that the bottleneck *T* extracts about the relevancy variable *Y* can only come from the source *X*. The solutions to (Equation 1) for varying λ trace the so-called *information curve*, i.e., the λ-parameterised curve
(2)$$\begin{array}{c} {\left(I_{\lambda }\left(X;T\right),I_{\lambda }\left(Y;T\right)\right)}_{\lambda \geq 0}\subseteq R^{2}, \end{array}$$
where $I_{\lambda }\left(X;T\right)$ and $I_{\lambda }\left(X;T\right)$ are defined by a bottleneck *T* of parameter λ (see the black curve in the first figure in Section 2 below). This curve indicates the informationally optimal bounds on the feasible trade-offs between relevancy I(Y;T) and complexity I(X;T) of the representation *T*. In this sense, the IB method provides a fundamental understanding of the *informationally optimal limits* of information-processing systems.

These limits are crucial for both understanding and building adaptive behaviour. For instance, choosing *X* to be an agent’s past and *Y* to be its future leads it to extract the most relevant features of its environment [3,4,5,6]. More generally, the IB point of view on modelling embodied agents’ representations has been leveraged for unifying efficient and predictive coding principles in theoretical neuroscience—at the level of single neurons [3,7,8,9] and neuronal populations [9,10,11,12,13]—but also for studying sensor evolution [14,15,16], the emergence of common concepts [17] and of spatial categories [18], the evolution of human language [19,20,21], or for implementing informationally efficient control in artificial agents [22,23,24]. This line of research brings increasing support to the hypothesis that, particularly for evolutionary reasons, biological agents are often poised close to optimality in the IB sense. It also provides a framework for both measuring and improving artificial agents’ performance.

However, one aspect of the IB framework conflicts with a crucial feature of real-world systems: the informationally optimal limits that it describes only consider a given representation *T taken in isolation* from any other one in the system. This point of view *a priori* disregards the *relationship between representations*, which is crucial in real-world information-processing systems. Thus, it is crucial to consider the following question: does the relationship between a set of internal representations $T_{1},\cdots ,T_{n}$ impact their individual information optimality? In this paper, we are mostly interested in a specific kind of relationship: when $T_{1},\cdots ,T_{n}$ are successively produced in this order, and each new $T_{i}$ builds on both the previous representation $T_{i-1}$ and new information from the fixed source *X* to extract information about the fixed relevancy *Y*. This scenario formalises the *incorporation of information into already learned representations*—as is the case in developmental learning, or, more generally, any kind of learning process that goes through identifiable successive steps.

More precisely, consider an informationally bounded agent that extracts information about a relevant variable *Y* within an environment *X*. If the agent is informationally optimal, given an affordable complexity cost $\lambda_{1}$, it must maximise the relevant information that it extracts from the environment—resulting in a bottleneck representation $T_{1}$, i.e., a solution to (Equation 1) with parameter $\lambda_{1}$. Then, assume that at, a later stage, the complexity cost that the agent can afford increases to $\lambda_{2}>\lambda_{1}$, while the goal is still to extract information about the same relevant feature *Y* within the same environment *X*. To keep being informationally optimal, the agent should thus update its representation so it becomes a bottleneck of parameter $\lambda_{2}$. Given this setting, the question we ask is: to which extent can the content learned into $T_{1}$ be leveraged for the production of $T_{2}$? It is indeed not intuitively clear that $T_{2}$ should keep all the information from $T_{1}$. An informal example is the fact that most pedagogical curricula teach knowledge via successive approximations, where, at a more advanced level, the content learned at the beginner level must sometimes be *unlearned* to successively proceed further, even though it was perfectly reasonable—in our language, informationally optimal—to deliver the first beginner sketch to students that would never progress to learn the expert level.

This question has been formalised, in the rate-distortion literature, with the notion of *successive refinement* (SR) [25,26,27,28,29], which, in short, refers to the situation where several-stage processing does not incur any loss of information optimality. More precisely, in the context outlined above, there is successive refinement if the processing cost of first producing a coarse bottleneck $T_{1}$ of parameter $\lambda_{1}$ and then refining it to a finer bottleneck $T_{2}$ of parameter $\lambda_{2}>\lambda_{1}$ is no larger than the processing cost of directly producing a bottleneck $T_{2}$ of parameter $\lambda_{2}$ without any intermediary bottleneck $T_{1}$ (see Section 2.1 and Section B.2 for formal definitions). The aim of this work is to push the understanding of successive refinement in the IB framework [30,31,32] further, as well as to expand the analysis to a *quantification* of the lack of SR, in cases where the latter does not hold exactly. We start by leveraging general results in existing IB literature [33,34] to prove that successive refinement always holds for jointly Gaussian (X,Y), and when *Y* is a deterministic function of *X*. However, it is seems crucial, for further progress on more general scenarios, to design specifically tailored mathematical and numerical tools. In this regard, we provide two main contributions.

First, we present a simple geometric characterisation of SR, in terms of convex hulls of the decoder symbol-wise conditional probabilities q(X|t), for *t* varying in the bottleneck alphabet T. This characterisation is proven in the discrete case under an additional but mild assumption of injectivity of the decoder q(X|T). This new point of view fits well with an ongoing convexity approach to the IB problem [35,36,37,38,39] and might thus help develop a new geometric perspective on the successive refinement of the IB. As an example, we use this geometric characterisation to prove that SR always holds for binary source *X* and binary relevancy *Y*. Moreover, this characterisation makes it straightforward to numerically assess, with a linear program checking convex hull inclusions, whether or not two discrete bottlenecks $T_{1}$ and $T_{2}$ achieve successive refinement. As we demonstrate with minimal numerical examples, this can help in investigating the SR structure of any given IB problem, i.e., how successive refinement depends on the particular combination of trade-off parameters $\lambda_{1}$ and $\lambda_{2}$.

Second, we soften [18] the traditional notion of successive refinement and study the *extent to which* several-stage processing incurs a loss of information optimality. More precisely, we propose to measure soft successive refinement with the *unique information* [40] (UI) that the coarser bottleneck $T_{1}$ holds about the source *X*, as compared to the finer one $T_{2}$. Explicitly, this UI is defined as the minimal value of $I_{q}\left(X;T_{1}|T_{2}\right)$ over all distributions $q:=q(X,T_{1},T_{2})$ whose marginals $q(X,T_{1})$ and $q(X,T_{2})$ coincide with the corresponding bottleneck distributions (see Section 3.1 for details). As a first exploration of soft SR’s qualitative features, we investigate the landscapes of unique information over trade-off parameters, for again some simple example distributions p(X,Y). These landscapes seem to unveil a rich structure, which was largely hidden by the traditional notion of SR, that only distinguished between SR being present or absent. Among the general features suggested by these experiments, the most significant are that (i) soft SR seems strongly influenced by the trajectories of the decoders $q_{\lambda }\left(X|T\right)$ over λ; (ii) the UI often goes through sharp variations at the bifurcations [41,42,43,44] undergone by the bottlenecks (in a fashion compatible with the presence of discontinuities of either the UI itself, or its differential, with regard to trade-off parameters); and (iii) the loss of information optimality seems always small—more precisely, the global bound on the UI was observed to be typically one or two orders of magnitude lower than the system’s globally processed information (see Section 3.2 for formal statements). These three conclusions are phenomenological and limited to our minimal examples, but they shed light on the kind of structure that can be investigated by further research. They also suggest the relevance that developing this theoretical framework might have for the scientific question that motivates it. In particular, the link with IB bifurcations and the overall small loss of information optimality would, if generalisable, have interesting consequences for the structure and efficiency of incremental learning.

As a side contribution, we draw along the paper formal equivalences between our framework and other notions proposed in the literature, thus making the formal framework also relevant to decision problems [40,45] and to the information-theoretic approach to deep learning [46]. This flexibility of interpretation stems from the fact that even though our formal framework crucially depends on the order of the bottleneck representations’ trade-off parameters, it does not depend on the order in which these representations are produced. Thus, a sequence of bottlenecks can be equally well interpreted as produced from coarsest to finest—as is the case for the information incorporation interpretation outlined above—or from finest to coarsest—as is the case in feed-forward processing. This conceptual unity sheds light on the common formal structure shared by these diverse phenomena.

In the next Section 1.2, we review related work. After having established notations and recalled some general notions in Section 1.3, we formally introduce the notion of the successive refinement of the IB in Section 2.1, where we also prove successive refinability in the case of Gaussian vectors and deterministic channel p(Y|X). We then present the convex hull characterisation in Section 2.2, before using it to prove successive refinement for the case of binary source and relevancy variables. The following Section 2.3 leverages the convex hull characterisation to gather some first insights from minimal experiments. These experiments suggest an intuition for defining soft successive refinement, which we formalise in Section 3.1 through a measure of unique information [40], where we provide theoretical motivations for our choice. This new measure is explored in Section 3.2 with additional numerical experiments that highlight the general features described above. The alternative interpretations of both exact and soft SR, in terms of decision problems and feed-forward deep neural networks, are developed in Section 4.1 and Section 4.2, respectively. We then describe the limitations and potential future work in Section 5, and conclude in Section 6.

### 1.2. Related Work

The notion of successive refinement has been long studied in the rate-distortion literature [25,26,27,28,29]. However, classic rate-distortion theory [47] usually considers distortion functions defined on the random variables’ *alphabets*, whereas the IB framework can be regarded as a rate-distortion problem only if one allows the distortion to be defined on the space of probability *distributions* [48]. Successive refinement thus needed to be adapted to the IB framework, which was achieved starting from various perspectives.

In [30,31], successive refinement is formulated within the IB framework. Then, Ref. [32] goes further by considering the informationally optimal limits of several-stage processing in general, without comparing it to single-stage processing. In both these works, the problem is initially defined in asymptotic coding terms, and only then given a single-letter characterisation. On the contrary, we will directly define successive refinement from a single-letter perspective. It turns out that our single-letter definition and the operational multi-letter definition from [30,31] are equivalent. The two latter works—as well as [32]—thus provide our single-letter definition with an operational interpretation that also formalises the intuition of an informationally optimal incorporation of information (see Proposition 1 and Section B.2).

Another notion named “successive refinement” as well can be found in [46]. This work, instead of modelling information incorporation, rather considers the successive processing of data along a feed-forward pipeline—which encompasses the example of deep neural networks. Fortunately, the “successive refinement” defined in [46] happens to encompass the notion we develop here; more precisely, in [46], the relevancy variable is allowed to vary across processing stages, but if we choose it to be always the same, then “successive refinement” as defined in [46] and “successive refinement” as defined here are formally equivalent (see Section 4.2). In other words, the situation considered in this paper is a particular case of [46], so our results, methods, and phenomenological insights are directly relevant to [46]. For instance, our proof of SR for binary *X* and *Y* (see Proposition 5) is a generalisation of Lemma 1 in [46], which proves SR when *X* is a Bernoulli variable of parameter $\frac{1}{2}$ and p(Y|X) is a binary symmetric channel.

More generally speaking, the link between successive refinement and the IB theory of deep learning [49,50,51,52,53,54,55,56] has been noted since the inception of the latter research agenda [49], and, besides in [46], it was also further developed in [57]. Section 4.2 makes clear in which sense our results are relevant to this line of research. In particular, our minimal experiments suggest (if they are scalable to the much richer deep learning setting) that trained deep neural networks should lie close to IB bifurcations: i.e., if *X* is the network’s input, *Y* the feature to be learnt and $L_{1},\cdots ,L_{n}$ the network’s successive layers, the points $\left(I\left(X;L_{i}\right),I\left(Y,L_{i}\right)\right)$ should lie close to points of the information curve corresponding to IB bifurcations. This feature was already suggested in [49,50], but for reasons not explicitly related to successive refinement. Note that while the phenomenon of IB bifurcations has been studied from a variety of perspectives (see, e.g., [41,42,43,44]), here, we adopt that of [43], which frames IB bifurcations as parameter values where the minimal number of symbols required to represent a bottleneck increases.

In [58], successive refinability is proved for discrete source *X* and relevancy Y=X. Our Proposition 3 generalises this result to either discrete or continuous source *X*, with relevancy *Y* being an arbitrary function of *X*, with a similar argument as that in [58].

In [33], links between the IB framework and renormalisation group theory are exhibited. Even though the questions addressed in the latter work are thus distinct from those addressed here, the Gaussian IB’s *semigroup structure* defined and proven in [33] implies the successive refinability of Gaussian vectors (see Proposition 2, and see Appendix 2 for more details on the semigroup structure). This generalises Lemma 3 in [46], which proves SR when *X* and *Y* are jointly Gaussian, but each one-dimensional (see Section 4.2 for the relevance of [46] to our framework).

The geometric approach in which we propose to study the successive refinement of the IB is closely related to the convexity approach to the IB [35,36,37,38,39], which frames the IB problem as that of finding the lower convex hull of a well-chosen function. This formulation happens to fit neatly with our convex hull characterisation of successive refinement; we use it to apply the characterisation to proving successive refinability in the case of a binary source and relevancy. Moreover, it is worth noting that our convex hull characterisation makes successive refinement tightly related to the notion of *input-degradedness* [59], through which additional operational interpretations can be given to successive refinement, particularly in terms of randomised games.

The loss of information optimality induced by several-stage processing has already been studied in [60] (see next paragraph), but a quantification of it based on *soft Markovianity* was, to the best of our knowledge, only considered in [18]. Here, we take inspiration in the latter work to quantify soft successive refinement, but we explicitly address the problem that joint distributions over distinct bottlenecks are not uniquely defined. This leads us to use the *unique information* defined in [40] within the context of partial information decomposition [61,62,63,64] as our measure of soft SR. This unique information has tight links with the Blackwell order [45,65], which allows us in Section 4.1 to provide a second alternative interpretation of (exact and soft) successive refinement in terms of decision problems.

Ref. [60] proves the near-successive refinability of rate-distortion problems when the distortion measure is the squared error. However, the latter work’s approach is different from ours in two respects. First, the distortion measures are different: in particular, as mentioned above, the IB distortion is defined over the space of probability distributions on symbols, unlike the squared error, which is defined on the space of symbols itself. Second, Ref. [60] quantifies the lack of SR as the respective differences between sequences of optimal rates (for given distortion sequences) of a several-stage processing system and the corresponding optimal rates (for the same distortions) of a single-stage processing system. Here, we quantify the lack of SR with a single quantity: the unique information defined by bottlenecks with different granularities. We are, at this stage, not aware of a link between this value of unique information and differences in one-stage and several-stage optimal rates.

### 1.3. Technical Preliminaries

In this section, we fix the notations and conventions that we will use along the paper and recall some general notions that we will need.

#### 1.3.1. Notations and Conventions

The random variables are denoted by capital letters, e.g., *X*, their alphabets by calligraphic ones, e.g., X, and their symbols by lower-case letters, e.g., *x*. Sometimes, we will mix upper- and lower-case notations to denote a family where some symbols vary, while others are fixed, e.g., $q\left(X|t\right):={\left(q(x|t)\right)}_{x\in X}\;$, or $q\left(x|T\right):={\left(q(x|t)\right)}_{t\in T}\;$. Throughout the whole paper, *X* is the fixed source and *Y* the fixed relevancy of the IB problem. The variable *T* defined by the solution q(T|X) to the primal IB problem (Equation 1) is called a *primal* bottleneck. We use the same symbol *T* for *Lagrangian* bottlenecks, i.e., variables defined by solutions q(T|X) to the Lagrangian bottleneck problem (see Equation (Equation 3) below). By “bottleneck” without further specification, we refer to either a primal or Lagrangian bottleneck. The fixed source-relevancy distribution is denoted p(X,Y), and any distribution involving at least one bottleneck is denoted with the letter *q*, e.g., q(X,Y,T). When it is necessary to make the trade-off parameter explicit, we index the corresponding objects by λ, e.g., $q_{\lambda }\left(T|X\right)$ or $I_{\lambda }\left(Y;T\right)$. Unless explicitly stated otherwise, the source *X*, relevancy *Y*, and any considered bottleneck *T* are defined as either all discrete or all continuous. Probability simplices, and sometimes some of their subsets are written using the generic symbol Δ; for instance, the source simplex is denoted by $\Delta_{X}$.

Without loss of generality, we always restrict *X*, *Y*, and the bottleneck *T* to their respective supports so that, in particular, all the conditional distributions are unambiguously well-defined, both in the discrete and the continuous case.

We will denote by $I_{Y}$ the function from $R_{+}$ to $R_{+}$ defined by $I_{Y}\left(\lambda \right):=I\left(Y;T\right)$, where *T* is a solution to the primal IB problem (Equation 1) for the parameter λ. The *information curve*, defined above in Equation (Equation 2), is thus also the graph of the function $I_{Y}$.

#### 1.3.2. General Facts and Notions

The following properties of the IB framework will be useful [35,37]:A bottleneck must saturate the information constraint, i.e., solutions *T* to (Equation 1) must satisfy $I_{\lambda }\left(X;T\right)=\lambda$. In other words, the primal trade-off parameter is the complexity cost of the corresponding bottleneck.The function $I_{Y}:\lambda \;\mapsto \;I_{\lambda }\left(T;Y\right)$ is constant for λ≥H(X). We will thus always assume, without loss of generality, that λ∈[0,H(X)].In the discrete case, choosing a bottleneck cardinality |T|=|X|+1 is enough to obtain optimal solutions. Thus, we always assume, without loss of generality, that |T|≤|X|+1, where |T|<|X|+1 might occur if needed to make *T* full support.

To compute bottleneck solutions, instead of directly solving the primal problem (Equation 1), following common practice, we will solve its Lagrangian relaxation [66]:(3)$$\begin{array}{c} \underset{\begin{array}{c} q(T|X)\;:\;T-X-Y, \end{array}}{arg min}\;I\left(X;T\right)\;-\;\beta \;I\left(Y;T\right), \end{array}$$
where the complexity-relevancy trade-off is now parameterised by β≥0, which corresponds to the inverse of the information curve’s slope [41]. As the information curve is known to be concave, the Lagrangian parameter β is an increasing function of the primal parameter λ=I(X;T). Moreover, we can, without loss of generality, assume that β≥1 [43]. (Note that when the information curve is not strictly concave, the Lagrangian formulation does not allow one to obtain all the solutions to the primal problem [39,67]. However, in our simple numerical experiments, we always obtained strictly concave information curves.)

We will also need the following concepts [43]:

**Definition** **1.***Let T be a (primal or Lagrangian) discrete bottleneck. The* effective cardinality *k=k(T) is the number of distinct pointwise conditional probabilities q(X|t) for varying t.*

**Definition** **2.***A discrete (primal or Lagrangian) bottleneck T is a* canonical bottleneck, *or is in* canonical form, *if all the pointwise conditional probabilities q(X|t) are distinct, i.e., equivalently, if |T|=k(T), where k(T) is the effective cardinality of T.*

Our definition of effective cardinality, even though slightly different from the original one in [43], is equivalent to the latter for Lagrangian bottlenecks. And, importantly, every (primal or Lagrangian) bottleneck can be reduced to its canonical form by merging the symbols with identical q(X|t) (see Section A.1 for more details). We will be particularly interested in the *change* of effective cardinality, which has been identified in [43] as characterising the bottleneck phase-transitions, or *bifurcations*.

In Figure 1, we present examples of bottleneck conditional distributions q(X|T), visualised as the family of points {q(X|t),t∈T} on the source simplex $\Delta_{X}$, where, here, |X|=3, and the bottleneck is computed with |T|=3 in both examples. However, in Figure 1 (left), there are only two distinct q(X|t), so there must be two equal pointwise probabilities $q(X|t_{1})$ and $q(X|t_{2})$; thus, k=2 and the canonical form of *T* is obtained by merging $t_{1}$ and $t_{2}$. On the contrary, in Figure 1 (right), there are three distinct q(X|t), so, here, k=3 and the bottleneck is already in canonical form.

Eventually, the notions of *consistency* and *extension* will be crucial to us.

**Definition** **3.**
*Let $A:=A_{1}\times \cdots \times A_{m}$ be a Cartesian product of (continuous or discrete) alphabets. For $C=\left\{c_{1},\cdots ,c_{r}\right\}\subseteq \left\{1,\cdots ,m\right\}$ a subset of coordinates, we write*

$$\begin{array}{c} \underset{c\in C}{\times }A_{c}:=A_{c_{1}}\times \cdots \times A_{c_{r}}. \end{array}$$

*For each 1≤i≤n, we consider a subset of coordinates $C_{i}$ and a probability distribution $q_{i}$ over $\times_{c\in C_{i}}A_{c}$. The distributions $q_{1},\cdots ,q_{n}$ are said to be* consistent *if, for every 1≤i,j≤n, the respective marginals of $q_{i}$ and $q_{j}$ on their common coordinates $\times_{c\in C_{i}\cap C_{j}}A_{c}$ are equal.*

For instance, if $T_{1}$ and $T_{2}$ are two bottlenecks, they define consistent distributions $q_{1}\left(X,Y,T_{1}\right)$ and $q_{2}\left(X,Y,T_{2}\right)$ because, by definition, their respective marginals on their common coordinates X×Y are $q_{1}\left(X,Y\right)=q_{2}\left(X,Y\right)=p\left(X,Y\right)$.

**Definition** **4.***Let $A:=A_{1}\times \cdots \times A_{m}$ be a Cartesian product of (continuous or discrete) alphabets, and $q_{1},\cdots ,q_{n}$ be consistent probability distributions over distinct but potentially overlapping coordinates of A. A distribution q over the whole A is called an* extension *of the family of distributions $\left\{q_{1},\cdots ,q_{n}\right\}$ if it is consistent with each $q_{i}$.*

Consider bottlenecks $T_{1},\cdots ,T_{n}$ of same source *X* and relevancy *Y* for resp. parameters $\lambda_{1},\cdots ,\lambda_{n}$. They define a consistent family of distributions $\left\{q_{\lambda_{i}}\left(X,T_{i}\right),\;1\leq i\leq n\right\}$. One of the central mathematical objects of this work is the set of their extensions into *joint* distributions $q(X,T_{1},\cdots ,T_{n})$:

**Notation** **1.**
*For given bottlenecks $T_{1},\cdots ,T_{n}$ of respective parameters $\lambda_{1},\cdots ,\lambda_{n}$, we denote by $\Delta_{\lambda_{1},\cdots ,\lambda_{n}}$ the set of extensions $q(X,T_{1},\cdots ,T_{n})$ of the family of distributions $\left\{q_{\lambda_{i}}\left(X,T_{i}\right),\;1\leq i\leq n\right\}$.*


In general, for a fixed family of bottlenecks, there is a multitude of possible ways to extend them into a joint distribution; indeed, $\Delta_{\lambda_{1},\cdots ,\lambda_{n}}$ traces a polytope on the simplex $\Delta_{X\times T_{1}\times \cdots \times T_{n}}$ of joint distributions (see Appendix A in [40]). This feature is the formal version of our previous statement that the IB framework does not entirely specify the *relationship* between representations $T_{1},\cdots ,T_{n}$: it only constrains it through the set $\Delta_{\lambda_{1},\cdots ,\lambda_{n}}$. Questions about possible relationships between IB representations are thus questions about properties of the set $\Delta_{\lambda_{1},\cdots ,\lambda_{n}}$.

## 2. Exact Successive Refinement of the IB

### 2.1. Formal Framework and First Results

Here, we formally describe, within the IB framework, the rate-distortion-theoretic notion of *successive refinement* (SR) [25,26,27,29]. We propose a purely single-letter definition (i.e., we only consider single source, relevancy, and bottleneck variables), which makes the presentation simpler but still conveys the intuition of information incorporation. After having presented the notion of SR in the IB framework, we describe its Markov chain characterisation (see Proposition 1), which mirrors the characterisation of SR for classic rate-distortion problems [26], and makes our formulation equivalent to previous multi-letter operational definitions, which also formalise the intuition of information incorporation [30,31,32]. We then leverage this characterisation to prove SR in the case of Gaussian vectors and deterministic channel p(Y|X).

Intuitively, there is successive refinement when a finer bottleneck $T_{2}$ does not discard any of the information extracted by a coarser bottleneck $T_{1}$. This can be imposed by requiring that $T_{2}=\left(T_{1},S_{2}\right)$ for some variable $S_{2}$, which encodes the “supplement” of information that “refines” $T_{1}$ into $T_{2}$. In the general case:

**Definition** **5.***Let $0<\lambda_{1}<\cdots <\lambda_{n}$, and a discrete or continuous p(X,Y) be given. There is* successive refinement *(SR) for parameters $\left(\lambda_{1},\cdots ,\lambda_{n}\right)$ if there exist variables $\left(T_{1},S_{2},S_{3},\cdots ,S_{n}\right)$ such that*

*$T_{1}$ is a bottleneck with parameter $\lambda_{1}$;*

*For every 2≤i≤n, the variable $T_{i}:=\left(T_{i-1},S_{i}\right)$ is a bottleneck with parameter $\lambda_{i}$.*



Note that even though it does not appear explicitly in this definition, the relevancy variable *Y* is indeed crucial to it, as it defines what a bottleneck is (see Equation (Equation 1)). If the conditions of Definition 5 hold, we will also say that the IB problem defined by p(X,Y) is $\left(\lambda_{1},\cdots ,\lambda_{n}\right)$-refinable. If bottlenecks $T_{1},\cdots ,T_{n}$ satisfy the definition’s conditions, we will say that they *achieve* successive refinement, or, simply, that there is successive refinement between these bottlenecks. If there is successive refinement for all combinations $0<\lambda_{1}<\cdots <\lambda_{n}$ of trade-off parameters, we will say that the corresponding IB problem is successively refinable. Eventually, when it will be needed in later sections to contrast this notion with that of soft successive refinement, we will refer to it as *exact* successive refinement.

For instance, let $0<\lambda_{1}<\lambda_{2}$ and X=Y={0,1}. We consider Y:=X⊕Z, where ⊕ denotes the modulo-2 addition, and *X* and *Z* are Bernouilli variables with parameters $\frac{1}{2}$ and *a*, respectively, for an arbitrary $0\leq a\leq \frac{1}{2}$. In this case, it is proven in Lemma 1 of [46] that, for well-chosen binary variables $S_{1}$ and $S_{2}$, we have that *X*, $S_{1}$, and $S_{2}$ are mutually independent, and the variables $X⊕S_{1}$ and $X⊕S_{1}⊕S_{2}$ are bottlenecks of resp. parameters $\lambda_{1}$ and $\lambda_{2}$. Moreover, using the independence of $S_{2}$ with $\left(X,X⊕S_{1}\right)$ and the assumed Markov chain $Y-X-X⊕S_{1}⊕S_{2}$, a straightforward computation shows that to get a bottleneck of parameter $\lambda_{2}$, the variable $X⊕S_{1}⊕S_{2}$ can be replaced by $\left(X⊕S_{1},S_{2}\right)$. Thus, here, the IB problem is $\left(\lambda_{1},\lambda_{2}\right)$-refinable, where successive refinement is achieved by $T_{1}:=X⊕S_{1}$ and $T_{2}=\left(T_{1},S_{2}\right)$.

It is helpful to visualise SR on the information plane, i.e., that on which lies the information curve. Indeed, successive refinement can be understood in terms of specific translations on the information plane: those resulting from concatenating an already existing variable $T_{i-1}$ with a new variable $S_{i}$—let us call them “accumulative translations” because they result from a processing that does not discard any of the information already collected. Let us focus on the case n=2 and first note that, whether or not $\left(T_{1},S_{2}\right)$ is a bottleneck, we have
$$\begin{array}{c} I\left(X;T_{1},S_{2}\right)=I\left(X;T_{1}\right)+I\left(X;S_{2}|T_{1}\right), \end{array}$$
and, similarly,
$$\begin{array}{c} I\left(Y;T_{1},S_{2}\right)=I\left(Y;T_{1}\right)+I\left(Y;S_{2}|T_{1}\right). \end{array}$$

In other words, the measure of both the complexity cost and relevance for $\left(T_{1},S_{2}\right)$ can be decomposed into the same measures first for $T_{1}$ and then for the “supplement” of information $S_{2}$, conditionally on the “already collected” information $T_{1}$. In Figure 2 (left and right), we first fix a coarse bottleneck $T_{1}$, understood here as a point $\left(I\left(X;T_{1}\right),I\left(Y;T_{1}\right)\right)$ on the information curve. Once $T_{1}$ is known, we supplement it with a new variable $S_{2}$, which incurs both an additional complexity cost $I(X;S_{2}|T_{1})$ and an additional relevant information gain $I(Y;S_{2}|T_{1})$. The question of successive refinement is that of whether the additional complexity cost can be leveraged enough for the resulting relevant information gain to take $\left(T_{1},S_{2}\right)$ “up to the information curve”, i.e., to be such that $(I\left(X;T_{1},S_{2}\right),I\left(Y;T_{1},S_{2}\right))$ is on the information curve. This is the case in Figure 2, right, and not the case in Figure 2, left. In short, there is successive refinement between two points on the information curve if and only if there exists an “accumulative translation” from the coarser one to the finer one.

Let us now describe a more formal characterisation, where point (ii) will mirror the characterisation of SR for classic rate-distortion problems [26].

**Proposition** **1.**
*Let $0<\lambda_{1}<\cdots <\lambda_{n}$. The following are equivalent:*

(i)
*There is successive refinement for parameters $\left(\lambda_{1},\cdots ,\lambda_{n}\right)$;*
(ii)
*There exist bottlenecks $T_{1},\cdots ,T_{n}$, of common source X and relevancy Y, with respective parameters $\lambda_{1},\cdots ,\lambda_{n}$, and an extension $q(X,T_{1},\cdots ,T_{n})$ of the $q_{i}:=q_{i}\left(X,T_{i}\right)$, such that, under q, we have the Markov chain*

(4)
$$\begin{array}{c} X-T_{n}-\cdots -T_{1}. \end{array}$$

(iii)
*There exist bottlenecks $T_{1},\cdots ,T_{n}$, of common source X and relevancy Y, with respective parameters $\lambda_{1},\cdots ,\lambda_{n}$, and an extension $q(Y,X,T_{1},\cdots ,T_{n})$ of the $q_{i}:=q_{i}\left(Y,X,T_{i}\right)$, such that, under q, we have the Markov chain*

(5)
$$\begin{array}{c} Y-X-T_{n}-\cdots -T_{1}. \end{array}$$




**Proof.** See Section B.1. It is relatively straightforward because we started directly from a single-letter definition. □

Proposition 1 was already known to be a characterisation of SR of the IB [30,31,32]. However, as the latter references start from an operational problem in terms of asymptotic rates and distortions for multi-letter systems, here, Proposition 1 shows that our single-letter Definition 5 is equivalent to the operational definitions in [30,31,32]. See Section B.2 for more details.

**Remark** **1.**
*Crucially, the order of the indexing in (Equation 4) and (Equation 5) depends only on the order of the trade-off parameters $\lambda_{1}<\cdots <\lambda_{n}$, and not on the order in which the bottlenecks $T_{i}$ are produced, which is just the interpretation we started from. In particular, Proposition 1 makes equally legitimate the interpretation of bottlenecks produced from the finest one to the coarsest one, each new bottleneck thus implementing a further coarsening of the source X. This alternative interpretation renders successive refinement relevant to feed-forward processing, including in particular the Blackwell order (see Section 4.1) and deep neural networks (see Section 4.2). For ease of presentation, though, we will stick to the information incorporation interpretation along most of the paper.*


Moreover, from Proposition 1, we can leverage existing IB literature to prove the successive refinability of two specific settings. (For an explicit definition of what we mean, in Proposition 2, by successive refinement in the case of the *Lagrangian* IB problem, see Section B.3.)

**Proposition** **2.**
*If X,Y are jointly Gaussian vectors, then the Lagrangian IB problem defined by p(X,Y) is $\left(\lambda_{1},\cdots ,\lambda_{n}\right)$-refinable for all $\lambda_{1}<\cdots <\lambda_{n}$.*


This result is a direct consequence of a property named a *semigroup structure*, and is proven for the Gaussian IB framework in [33], which relates the latter framework with renormalisation group theory. The semigroup structure denotes, in short, the situation where iterating the operation of coarse graining a variable by computing a bottleneck—where, at each iteration, the previous bottleneck becomes the source of the next IB problem—still outputs a bottleneck for the original problem. This semigroup structure is a stronger property than successive refinement and, as it is satisfied in the Gaussian case, this implies the successive refinability of Gaussian vectors (see Section B.3 for more details). Beyond Proposition 2, this relationship between successive refinement and the semigroup structure hints at potentially interesting links between the composition of coarse-graining operators and successive refinement. In this respect, note that our numerical results below (see Section 2.3 and Section 3.2) suggest that, for non-Gaussian vectors, successive refinement does not always hold and thus, *a fortiori*, that the semigroup structure might not always be satisfied in the IB framework—or at least not perfectly.

Eventually, in the case of deterministic channel p(Y|X), an explicit solution to the IB problem (Equation 1) is known [34]: T=Y with probability α, and T=e with probability 1−α, for *e* a dummy symbol, and some well-chosen 0<α<1. This specific solution allows one to address successive refinement for the deterministic case:

**Proposition** **3.**
*Let X be a discrete or continuous variable, and Y be a deterministic function of X. Then, the IB problem defined by p(X,Y) is successively refinable for all trade-off parameters $\lambda_{1}<\cdots <\lambda_{n}$.*


**Proof.** See Section B.4. A proof was already proposed, from an asymptotic coding perspective, for discrete *X* and Y=X, in [58]. We use a similar argument here. □

Note, though, that the solution used here to prove successive refinement is, as noted in [34], not very interesting: it is nothing more than an increasingly noisy version of *Y*. It is not clear whether or not there exists more interesting bottleneck solutions in the deterministic case, and if so, whether these other solutions are successively refinable. Proposition 3 will in any case be useful for our own purposes: we will use it to set aside the deterministic case in the proof of SR for binary *X* and *Y* (Proposition 5 below).

Until now, we used existing results from the IB literature that, even though not originally aimed at it, happen to yield interesting consequences for the problem of the successive refinement of the IB. However, it seems crucial, for further progress on the latter topic, to design specifically tailored mathematical and numerical tools. This is the purpose of the following sections of this paper; in particular, in the next section, we present a simple geometric characterisation of the IB’s successive refinability.

### 2.2. The Convex Hull Characterisation and the Case |X|=|Y|=2

In this section, we present our convex hull characterisation of successive refinement. We then show its relevance both to numerical computations—thanks to a linear program for checking the condition—and to proving new mathematical results—which we exemplify by proving, thanks to this new characterisation, the successive refinability of binary variables. Here, as in our subsequent numerical experiments in Section 2.3, we will focus on discrete variables and n=2 processing stages, even though our results are thought of as a first step towards a generalisation to continuous variables and an arbitrary number of processing stages.

The convexity approach that we propose hinges upon changing the perspective on the IB problem (Equation 1) from an optimisation over the encoder channels q(T|X) to an optimisation over the *decoder* channels q(X|T); indeed, (Equation 1) can be equivalently presented as the “reversed” optimisation problem
(6)$$\begin{array}{c} \underset{\begin{array}{c} (q(T),q(X|T))\;:\\ \sum_{t}q\left(t\right)q\left(X|t\right)=p\left(X\right)\\ T-X-Y,\;I(X;T)\leq \lambda \end{array}}{arg max}\;I\left(Y;T\right). \end{array}$$

Formulations (Equation 1) and (Equation 6) yield the same solutions because, through the Markov chain T−X−Y, the joint distribution q(X,Y,T) is equivalently determined by specifying some q(T|X) or specifying some pair $\left(q(T),q(X|T)\right)$ that satisfies the consistency condition $\sum_{t}q\left(t\right)q\left(X|t\right)=p\left(X\right)$. This condition says that the source distribution p(X) must be retrievable as a convex combination of the q(X|t), where the weights are given by the q(t).

Moreover, this formulation leads to a crucial intuition concerning the relationship between successive refinement and the set $H_{T}:=\mathrm{Hull}\left\{q\left(X|t\right),t\in T\right\}$, where, for a set $E\subseteq R^{n}$, we denote by Hull(E) the convex hull of *E*, i.e., the set of points obtained as convex combinations of points in *E*. First, note that, for a bottleneck *T*, the set $H_{T}$ is reduced to a single point if and only if *T* is independent from the source *X*. Conversely, $H_{T}$ coincides with the whole source simplex $\Delta_{X}$ if and only if *T* captures all the information from the source, i.e., if I(X;T)=H(X). Generalising these extreme cases suggests the intuition that $H_{T}$ describes the *information content* held by the bottleneck *T* about the source *X*. Now, let us recall that successive refinement from a coarse bottleneck $T_{1}$ to a finer bottleneck $T_{2}$ means intuitively that $T_{2}$ can be obtained without discarding any of the information extracted by $T_{1}$ about the source *X*; in other words, that the information content of $T_{1}$ about the source *Xis included* in that of $T_{2}$. Combining this latter intuition with the one about $H_{T}$ being the information content of a bottleneck *T* suggests the following characterisation of successive refinement:(7)$$\begin{array}{c} \mathrm{Hull}\left\{q\left(X|t_{1}\right),t_{1}\in T_{1}\right\}\subseteq \mathrm{Hull}\left\{q\left(X|t_{2}\right),t_{2}\in T_{2}\right\}, \end{array}$$
where $T_{1}$ and $T_{2}$ are bottlenecks of parameters $\lambda_{1}<\lambda_{2}$, respectively. This condition is visualised in Figure 3. The characterisation indeed holds, at least for the discrete case and under a mild assumption of injectivity of the finer bottleneck’s decoder:

**Proposition** **4.**
*Let $0<\lambda_{1}<\lambda_{2}$, and assume that p(X,Y) is discrete.*

*If there is successive refinement for parameters $\left(\lambda_{1},\lambda_{2}\right)$, then there exist bottlenecks $T_{1},T_{2}$ of parameters $\lambda_{1},\lambda_{2}$, respectively, such that the convex hull condition (Equation 7) is satisfied.*

*Conversely, if there exist bottlenecks $T_{1},T_{2}$ of parameters $\lambda_{1},\lambda_{2}$, respectively, such that the convex hull condition (Equation 7) holds and such that the decoder $q(X|T_{2})$, seen as a probability transition matrix, is injective, then there is successive refinement for parameters $\left(\lambda_{1},\lambda_{2}\right)$. Moreover in this latter case, if $T_{1}$, $T_{2}$ are bottlenecks that achieve successive refinement, the extension $\tilde{q}\left(X,T_{1},T_{2}\right)$ of $q(X,T_{1})$ and $q(X,T_{2})$ such that $X-T_{2}-T_{1}$ holds is uniquely defined.*


**Proof.** See Section B.5. The idea consists in translating the Markov chain characterisation $X-T_{2}-T_{1}$ into the convex hull condition (Equation 7). The direct sense is straightforward. For the converse direction, observe that, even though as soon as (Equation 7) is satisfied it provides a joint distribution $\tilde{q}\left(X,T_{1},T_{2}\right)$ that satisfies the Markov chain $X-T_{2}-T_{1}$, it is not clear whether this distribution is consistent with $q(X,T_{1})$. The potential problem stems from the fact that $\tilde{q}$ must be such that the channel $\tilde{q}\left(T_{2}|T_{1}\right)$ maps the marginal $q(T_{1})$ to the marginal $q(T_{2})$. The injectivity assumption, however, provides a sufficient condition for it to be the case. This assumption happens to also imply the uniqueness of the extension, among all those that satisfy the Markov chain $X-T_{2}-T_{1}$. □

Even though the injectivity assumption might seem restrictive, in practice, in our numerical experiments below (see Section 2.3 and Section 3.2), we always found that the decoder channel $q(X|T_{2})$ could be chosen as injective by reducing it to its effective cardinality (see Section 1.3)—a process that leaves the convex hull condition (Equation 7) unchanged because it leaves the points $q(X|t_{2})$ unchanged. See also Appendix D for a conjecture that, if true, would simplify our convex hull characterisation in the case of a strictly concave information curve.

**Remark** **2.***The convex hull condition happens to be equivalent to the* input-degradedness *pre-order on channels (see Proposition 1 in [59]). Even though we will not develop this point further when considering alternative interpretations of SR (Section 4), it is worth noting that, through input-degradedness, SR can be given additional operational interpretations, particularly in terms of randomised games (see Section IV-C in [59]).*

Our new characterisation provides a simple way of checking whether or not two bottlenecks $T_{1}$ and $T_{2}$ achieve SR. Recall that the Markov chain characterisation (Proposition 1, point (ii)) shows that SR is a feature of the space $\Delta_{q_{1},q_{2}}$ of all extensions $q(X,T_{1},T_{2})$ of individual bottleneck distributions $q_{1}\left(X,T_{1}\right)$ and $q_{2}\left(X,T_{2}\right)$. While this set might, *a priori*, be difficult to study directly, our characterisation (Equation 7) reduces the problem to a simple geometric property relating only two explicitly given conditional distributions: $q(X|T_{1})$ and $q(X|T_{2})$. Moreover, note that (Equation 7) is equivalent to
$$\begin{array}{c} \forall t_{1}\in T_{1},\;q\left(X|t_{1}\right)\in \mathrm{Hull}\left\{q\left(X|t_{2}\right),t_{2}\in T_{2}\right\}, \end{array}$$
and that checking whether a point is in the convex hull of a finite set of other points can be cast as a linear programming problem [68]. As a consequence, one can bound the time complexity of checking condition (Equation 7) as O(|X|K), where *K* is the time complexity bound of a linear program with 2|X|+2 variables and 3|X|+2 constraints. As a consequence, using the bound on *K* proved in [69], the time complexity of checking (Equation 7) is no worse that $\tilde{O}{\left(|X\right|}^{\omega +1}\log\left(\frac{\left|X\right|}{\delta }\right))$, where ω≈2.38 corresponds to the complexity of matrix multiplication, δ is the relative accuracy, and the $\tilde{O}\left(\cdot \right)$ notation hides polylogarithmic factors (see Section B.6 for details).

We deem this convex hull characterisation to be important for theory as well. Indeed, it reduces the question of successive refinement to a question about the structure of the trajectories, on the source probability simplex $\Delta_{X}$, of the points $q_{\lambda }\left(X|t\right)$ for varying λ. Thus, any theoretical progress on the description of these bottleneck trajectories might lead to theoretical progress on the side of successive refinement. As a first step in this direction, we show that this geometric point of view helps to solve the question of SR in the case of a binary source and relevancy (This result generalises the already known fact that there is always successive refinement when *X* is a Bernoulli variable of parameter $\frac{1}{2}$ and p(Y|X) is a binary symmetric channel (see Lemma 1 in [46] and see Section 4.2 for explanations on why the latter work’s framework encompasses ours). Moreover, a potential generalisation of our result to an arbitrary number of processing stages is left to future work).

**Proposition** **5.**
*If |X|=|Y|=2, then, for any discrete distribution p(X,Y) and any trade-off parameters $\lambda_{1}<\lambda_{2}$, the IB problem defined by p(X,Y) is $\left(\lambda_{1},\lambda_{2}\right)$-successively refinable.*


**Proof.** Let us here outline the proof presented in Section B.7. The case of deterministic p(Y|X) was already dealt with in Proposition 3, so we can assume that p(Y|X) is not deterministic. In this case, the IB problem with |X|=|Y|=2 and n=2 has been extensively studied in [35]. In short, the latter approach leverages the fact that a pair (q(T),q(X|T)) is a solution to the IB problem (Equation 6) if the convex combination of the points $F_{\beta }\left(q\left(X|t\right)\right)$, with weights given by q(T), achieves the lower convex envelope of the function $F_{\beta }$, where $F_{\beta }$ is a well-chosen function on the source simplex $\Delta_{X}$ and β is the information curve’s inverse slope. This work, along with considerations from [37], which uses the same convexity approach, yields in particular that (i) the points $q_{\beta }\left(X|t\right)$ are the extreme points of a non-empty open segment uniquely defined by β, and (ii) this latter segments grows as a function of the inverse slope β and thus, by concavity, as a function of λ. This implies that the convex hull condition is always satisfied for $\lambda_{1}<\lambda_{2}$. As point (i) also implies that, here, $q_{\lambda_{2}}\left(X|T_{2}\right)$ must be injective, Theorem 4 allows us to conclude the successive refinability for n=2 processing stages. □

The proof of Proposition 5 exemplifies how our convex hull characterisation interlocks well with the convexity approach to the IB [35,36,37,38,39]. In this sense, our characterisation brings a new theoretical tool to the study of the successive refinement of the IB.

### 2.3. Numerical Results on Minimal Examples

In this section, we leverage our new convex hull characterisation to investigate successive refinement on minimal numerical examples, i.e., with discrete and low-cardinality distributions p(X,Y). Our experiments suggest that, in general, successive refinement does not always hold exactly. However, they also highlight two other features: first, it seems that successive refinement is often shaped by IB bifurcations [41,42,43,44]. Second, even though successive refinement is often not satisfied exactly, visualisations suggest that it is often “close” to being satisfied. The formalisation of this latter intuition will be the topic of the next section.

We consider the Lagrangian form (Equation 3) of the IB problem (see Section 1.3). We compute solutions to it with the Blahut–Arimoto (BA) algorithm [1], combined with reverse deterministic annealing [19,70], starting from β≈∞ (i.e., in practice, β≫1) at the IB solution T=X (we noticed that regular deterministic annealing sometimes yielded sub-optimal solutions because they followed sub-optimal branches at IB bifurcations [1,71], which was not the case for reverse annealing). We always obtained that I(X;T) was a strictly increasing function of the Lagrangian parameter β, so it makes sense to index the solutions by λ=I(X;T) rather than β; for instance, in this section and Section 3.2, we will write $q_{\lambda }\left(T|X\right)$ for our algorithm’s output for a β such that I(X;T)=λ.

In all our numerical experiments, after reducing a bottleneck *T* to its canonical form (see Section 1.3), the decoder channel $q_{\lambda }\left(X|T\right)$ was injective. Therefore, thanks to Theorem 4, the convex hull condition (Equation 7) being satisfied here does imply successive refinement. In the remainder of the paper, we will thus use the convex hull condition as a proxy for numerically assessing successive refinement (see Appendix D for more details on what we mean by “proxy”). This condition can be investigated in two ways. First, for two distinct trade-off parameters $\lambda_{1}<\lambda_{2}$, we can compute whether the convex hull condition (Equation 7) holds or not with the linear program described in Section B.6. Second, for |X|≤3, we can visualise the whole trajectories, for varying λ, of the points $q_{\lambda }\left(X|t\right)$ on the source simplex $\Delta_{X}$. As we will see, this yields interesting qualitative insights.

As a sanity check for our algorithm, we compute bottleneck solutions for binary *X* and *Y*, which we proved in Proposition 5 to be successively refinable for all trade-off parameters. We used the linear program to check the convex hull condition numerically for all pairs $\lambda_{1}<\lambda_{2}$ and for distributions p(X,Y) uniformly sampled on the joint probability simplex $\Delta_{X\times Y}$. We find that the convex hull condition is indeed always numerically satisfied.

Then, we study the case |X|=|Y|=3, once again uniformly sampling example distributions p(X,Y) on $\Delta_{X\times Y}$. Figure 4, Figure 5 and Figure 6 show, for representative examples, visualisations of the trajectories over λ of the $q_{\lambda }\left(X|t\right)$ (left)—which we will refer to as the *bottleneck trajectories*—along with the corresponding computations of the convex hull condition as a function of $\lambda_{1}$ and $\lambda_{2}\geq \lambda_{1}$ (right)—which we will refer to as the *SR patterns* (The correspodning p(Y|X) are plotted in Appendix E, and p(X) is shown in Figure 4, Figure 5 and Figure 6 (left). The explicit p(X,Y) corresponding to each of these paper’s figures can be found at: https://gitlab.com/uh-adapsys/successive-refinement-ib/(accessed on 12 September 2023).

Let us first give a general description of the bottleneck trajectories. For λ≈0, the $q_{\lambda }\left(X|t\right)$ all coincide with the source distribution p(X). This should be the case, as, for 0=λ=I(X;T), the bottleneck *T* is independent of *X*. Then, when λ increases, the trajectories seem piecewise continuous, where each discontinuity corresponds to a symbol split, i.e., a change in effective cardinality (see Section 1.3). We mark with a cross, for each t∈T, the $q_{\lambda }\left(X|t\right)=q_{\lambda_{c}}\left(X|t\right)$ located just before such a change in effective cardinality.

In the examples of Figure 4, Figure 5 and Figure 6, as |X|=3, there are two symbol splits, corresponding to that from one to two and two to three symbols, respectively. Eventually, for large λ, the last continuous segment of bottleneck trajectories corresponds to effective cardinality $k(T_{\lambda })=|X|$, and, for the maximal λ, each corner of the source simplex $\Delta_{X}$ is reached by q(X|t) for some t∈T. This means that for maximum λ, there is a deterministic bijective relationship between *T* and *X*. The latter is expected: for maximum λ, bottlenecks are minimal sufficient statistics of *X* for *Y* [72]; where for p(X,Y) sampled uniformly on the simplex, these minimal sufficient statistics are, with probability 1, just permutations of *X*.

**Definition** **6.***In the following, we refer to the piece of trajectory where the bottleneck’s effective cardinality $k=k(T_{\lambda })$ is equal to the integer i as the*“segment k=i”, *i.e., it is the segment where $q_{\lambda }\left(X|T\right)$ corresponds to exactly i distinct points on the source simplex $\Delta_{X}$; for instance, in Figure 4, the segment k=2 corresponds to the first piece of trajectory spanning colors from dark blue to cyan.*

**Notation** **2.**
*We denote by $\lambda_{c}\left(i\right)$ the trade-off parameter’s critical value corresponding to the i-th change in effective cardinality, i.e., the symbol split from i to i+1 symbols. Here, we will only need to consider the critical values $\lambda_{c}\left(1\right)=0$ and $\lambda_{c}\left(2\right)$, corresponding to the splits from one to two and two to three symbols, respectively.*


Let us now come back to the question of successive refinement: for which parameters $\lambda_{1}<\lambda_{2}$ is the convex hull condition satisfied? The right-hand sides of Figure 4, Figure 5 and Figure 6 provide the answers corresponding to trajectories on the respective left-hand sides—where blue and red mean that the condition is and is not satisfied, respectively. Moreover, we highlight with dashed white vertical and horizontal lines the critical parameter values $\lambda_{1}=\lambda_{c}\left(i\right)$ and $\lambda_{2}=\lambda_{c}\left(i\right)$, respectively, at which the symbol split occurs (see Section B.8 for details on the computation of these symbols splits). Note that we always have $\lambda_{c}\left(1\right)\approx 0$, which is expected, as a bottleneck *T* corresponding to some λ=I(X;T)>0 must necessarily define at least two distinct $q_{\lambda }\left(X|t\right)$.

First, in these examples as in most non-reported examples, the convex hull condition (right) breaks as long as $\lambda_{2}<\lambda_{c}\left(2\right)$, i.e., as long as the finer bottleneck’s effective cardinality is at most k=2. This can also be read from the bottleneck trajectories (left): if the condition was satisfied for all $\lambda_{1}<\lambda_{2}<\lambda_{c}\left(2\right)$, for instance, then the segment k=2 would be a line segment. This is clearly not the case in Figure 4 and Figure 6, and even though visually it virtually seems to be the case in Figure 5, the segment k=2 happens to be very slightly curved, which is enough to break the convex hull condition. In other words, for $\lambda_{1}<\lambda_{2}<\lambda_{c}\left(i\right)$, several-stage processing seems to induce, in these examples, a nonzero loss of information optimality.

Then, for $\lambda_{2}>\lambda_{c}\left(2\right)$, even though there is no single general pattern, the trajectory’s structure at the bifurcation seems to impact successive refinement. Indeed, at the bifurcation at $\lambda_{c}\left(2\right)$, the set $\mathrm{Hull}\{q_{\lambda_{2}}\left(X|t\right),t\in T\}$ opens up along a new, third dimension, and keeps widening when $\lambda_{2}$ increases. This allows it to (gradually in Figure 4 and Figure 6, or virtually straight away in Figure 5) encompass the segment k=2 because it “overcomes” the curvature of this piece of trajectory. For instance, in Figure 4, because the segment k=2 is strongly curved, the convex hull condition gets satisfied for all $\lambda_{1}<\lambda_{c}\left(2\right)$ only if $\lambda_{2}$ is significantly larger than $\lambda_{c}\left(2\right)$. On the contrary, because in Figure 5, the segment k=2 is virtually not curved, it is almost as soon as $\lambda_{2}>\lambda_{c}\left(2\right)$ that the convex hull condition is satisfied for all $\lambda_{1}<\lambda_{c}\left(2\right)$.

In Figure 6, the lack of successive refinement for $\lambda_{2}>\lambda_{c}\left(2\right)$ does not seem to be due to the same phenomenon as the one just described. Generally speaking, we observed a whole variety of SR patterns (see Appendix F for more examples), and our aim here is not to try to interpret all of them. However, despite this diversity, the SR patterns that we studied typically shared a common qualitative feature: the bifurcation structure of the bottleneck trajectories seemingly participates in shaping these SR patterns. Mostly, it seems typically necessary, for SR to hold, that the larger parameter $\lambda_{2}$ has crossed the bifurcation value $\lambda_{c}\left(2\right)$, because the non-zero curvature of the segment k=2 can only be “overcome” by opening the set $\mathrm{Hull}\{q_{\lambda_{2}}\left(X|t\right),t\in T\}$ along a new dimension, through the symbol split at $\lambda_{2}=\lambda_{c}\left(2\right)$. This phenomenon will be explored in more details in Section 3.2.

Besides this relationship between SR and the structure of bottleneck bifurcations, this numerical study suggests a generalisation of the notion of successive refinement. Indeed, in Figure 5 for instance, even though the right-hand side asserts that successive refinement does not hold for $\lambda_{1}<\lambda_{2}<\lambda_{c}\left(2\right)$, the virtually linear piece of trajectory on the left-hand side suggests that this is “almost” the case. In the next section, we formalise this intuition.

## 3. Soft Successive Refinement of the IB

The minimal experiments from Section 2.3 suggest the intuition that even though successive refinement might not always hold exactly, when broken, it might still be “close” to being satisfied. More generally speaking, let us recall that we are trying here to understand the informationally optimal limits of several-stage information processing. As our numerical experiments suggest that the IB problem is not always successively refinable, it is desirable to *quantify* the lack of successive refinement—i.e., the lack of informational optimality induced by several-stage processing. These considerations lead to the notion of *soft successive refinement* [18], which we define and motivate in this section. As we will see, this generalisation of exact SR does not depend on the specific structure of the IB setting; rather, it can also be used as a generalisation of exact SR for *any* rate-distortion scenario.

### 3.1. Formalism

Let us first focus on the case n=2: we thus want to quantify the amount of information captured by a coarse bottleneck $T_{1}$ and then discarded by a finer bottleneck $T_{2}$. Let us recall that, from Proposition 1, bottlenecks $T_{1}$ and $T_{2}$ achieve successive refinement if there exists an extension $q(X,T_{1},T_{2})$ of $q_{1}\left(X,T_{1}\right)$ and $q_{2}\left(X,T_{2}\right)$ such that, under *q*, we have the Markov chain $X-T_{2}-T_{1}$, which is equivalent to $I_{q}\left(X;T_{1}|T_{2}\right)=0$. It thus seems natural to quantify soft successive refinement with the conditional mutual information $I_{q}\left(X;T_{1}|T_{2}\right)$. However, the IB method does not entirely define the relationship between distinct bottlenecks; formally, there is a whole polytope $\Delta_{q_{1},q_{2}}\subseteq \Delta_{X\times T_{1}\times T_{2}}$ of possible extensions $q(X,T_{1},T_{2})$ of $q_{1}\left(X,T_{1}\right)$ and $q_{2}\left(X,T_{2}\right)$ (see Section 1.3). Among these possible extensions, it seems natural to search for those that minimise the violation of the SR condition $I_{q}\left(X;T_{1}|T_{2}\right)=0$. This leads us to use the *unique information* [40]
(8)$$\begin{array}{c} UI\left(X:T_{1}\setminus T_{2}\right):=\min_{q\in \Delta_{q_{1},q_{2}}}I_{q}\left(X;T_{1}|T_{2}\right). \end{array}$$

This quantity was already defined in [40] in the context of partial information decomposition [61,62,63,64], and it happens to be relevant to us for several reasons.

First of all, it depends only on the distributions $q_{1}\left(X,T_{1}\right)$ and $q_{2}\left(X,T_{2}\right)$, which are indeed the only distributions provided by the IB framework. Second, from Proposition 1, there is successive refinement if and only if there are two bottlenecks $T_{1}$ and $T_{2}$ such that $UI_{q_{1},q_{2}}\left(X:T_{1}\setminus T_{2}\right)=0$. Third, it is thoroughly argued in [40] that (Equation 8) is a good measure of the information that only $T_{1}$, and not $T_{2}$, has about *X*, which is an interpretation that coincides neatly with the intuition that we want to operationalise here. Eventually, Proposition 6 below, which first requires some definitions, provides an information-geometric justification.

**Definition** **7.**
*For Δ a probability simplex and $E_{1},E_{2}\subseteq \Delta$, we define*

$$\begin{array}{c} D_{KL}(E_{1}\left|\right|E_{2}):=\underset{r_{1}\in E_{1},\;r_{2}\in E_{2}}{\inf}D_{KL}(r_{1}\left|\right|r_{2}), \end{array}$$

*where $D_{KL}$ is the Kullback–Leibler divergence: $D_{KL}(r_{1}\left|\right|r_{2}):=\sum_{a\in A}r_{1}\left(a\right)\log\left(\frac{r_{1}\left(a\right)}{r_{2}\left(a\right)}\right)$, if the probability distributions $r_{1}$ and $r_{2}$ are defined on the discrete alphabet A.*


**Definition** **8.***The* successive refinement set *$\Delta_{SR,n}\subseteq \Delta_{X\times T_{1}\times \cdots \times T_{n}}$ is the set of distributions r on $X\times T_{1}\times \cdots \times T_{n}$ such that, under r, the Markov chain $X-T_{n}-\cdots -T_{1}$ holds.*

Note that $\Delta_{SR,n}$ does not require its elements to be extensions of any fixed bottleneck distributions $q_{i}\left(X,T_{i}\right)$ but imposes the Markov chain that characterises SR (see Proposition 1). SR is achieved for bottlenecks $q_{1}\left(X,T_{1}\right),\cdots ,q_{n}\left(X,T_{n}\right)$ if and only if the successive refinement set $\Delta_{SR,n}$ and the extension set $\Delta_{q_{1},\cdots ,q_{n}}$ share a common distribution $q\in \Delta_{SR,n}\cap \Delta_{q_{1},\cdots ,q_{n}}$. In general (for n=2), the following proposition can easily be derived:

**Proposition** **6.**
*For fixed distributions $q_{1}=q_{1}\left(X,T_{1}\right)$, $q_{2}=q_{2}\left(X,T_{2}\right)$, we have*

(9)
$$\begin{array}{c} UI\left(X:T_{1}\setminus T_{2}\right)=D_{KL}(\Delta_{q_{1},q_{2}}\left|\right|\Delta_{SR,2}). \end{array}$$



**Proof.** See Section C.1. □

In this sense, $UI(X:T_{1}\setminus T_{2})$ quantifies “how far” the joint distributions extending the bottlenecks $T_{1}$ and $T_{2}$ are from making the successive refinement condition $X-T_{2}-T_{1}$ hold true, where the “distance” is understood as a minimised Kullback–Leibler divergence.

Our new measure of soft SR is continuous:

**Proposition** **7**([73], Property P.7). *The unique information $UI(X:T_{1}\setminus T_{2})$ is a continuous function of the probabilities $q_{1}\left(X,T_{1}\right)$ and $q_{2}\left(X,T_{2}\right)$.*

**Remark** **3.**
*In particular, if $UI(X:T_{1}\setminus T_{2})$ has a discontinuity as a function of the parameter $\lambda_{1}$ or $\lambda_{2}$, which define the bottleneck distribution $q_{\lambda_{1}}\left(X,T_{1}\right)$ or $q_{\lambda_{2}}\left(X,T_{2}\right)$, respectively, then this can only be a consequence of a discontinuity of the probability $q_{\lambda_{1}}\left(X,T_{1}\right)$ as a function of $\lambda_{1}$ or $q_{\lambda_{2}}\left(X,T_{2}\right)$ as a function of $\lambda_{2}$, itself, respectively. This consideration will be useful for analysing our numerical experiments in Section 3.2.*


Moreover, the formulation (Equation 9) of unique information suggests a natural generalisation to an arbitrary number of processing stages:

**Definition** **9.***Let $T_{1},\cdots ,T_{n}$ be bottlenecks with respective parameters $\lambda_{1}<\cdots <\lambda_{n}$, and $q_{i}\left(X,T_{i}\right)$ their respective individual distributions. One can quantify* soft successive refinement, *or, equivalently, the* lack of successive refinement, *through the divergence $D_{KL}(\Delta_{q_{1},\cdots ,q_{n}}\left|\right|\Delta_{SR,n})$.*

While [74] proposes a provably convergent algorithm to compute $UI(X:T_{1}\setminus T_{2})$, to the best of our knowledge, there currently exists no provably convergent algorithm to compute $D_{KL}(\Delta_{q_{1},\cdots ,q_{n}}\left|\right|\Delta_{SR,n})$ for n>2. Our numerical investigations (see Section 3.2) will stick to the case n=2, but this generalisation makes soft SR in particular, at least conceptually for now, more relevant to deep learning (see Section 4.2).

For the sake of completeness, let us point out that for each λ, there is a whole set of solutions $q_{\lambda }\left(T|X\right)$—or, equivalently, $q_{\lambda }\left(X,T\right)$—to the IB problem (Equation 1). Thus, the unique information, which is defined as a function of specific bottleneck distributions $q_{1}\left(X,T_{1}\right)$ and $q_{2}\left(X,T_{2}\right)$, could *a priori* not be uniquely defined by the corresponding trade-off parameters $\lambda_{1}$ and $\lambda_{2}$. This subtlety is further explained in Appendix D, where we also formulate a conjecture that would prove that, at least in the case of a strictly concave information curve, the trade-off parameters do uniquely define the unique information.

### 3.2. Numerical Results on Minimal Examples

A provably convergent algorithm that computes, in the discrete case, the unique information (Equation 8), was provided in [74]. In this section, we use the authors’ implementation of this algorithm (https://github.com/infodeco/computeUI, accessed on 12 September 2023) to qualitatively investigate, on minimal examples, the landscapes of unique information (UI) and their relationship to the bottleneck trajectories on the simplex.

In Figure 7, Figure 8 and Figure 9 (left), we plot again the same bottlenecks trajectories as in Figure 4, Figure 5 and Figure 6 (left), but compare them this time with the unique information $UI(X:T_{1}\setminus T_{2})$, plotted as a function of $\lambda_{1}$ and $\lambda_{2}$ (right). We also plot, in Figure 10, Figure 11 and Figure 12, some representative examples of the exact SR patterns (left) and UI landscapes (right) for slightly larger source and relevancy cardinalities, where p(X,Y) is, as above, uniformly sampled — the explicit distributions p(X,Y) corresponding to Figure 10, Figure 11 and Figure 12 can be found at https://gitlab.com/uh-adapsys/successive-refinement-ib/. (see Appendix F for additional examples of comparison of the UI landscapes with bottleneck trajectories, and with the exact SR patterns.) Once again, we highlight with dashed white vertical and horizontal lines the critical parameter values $\lambda_{1}=\lambda_{c}\left(i\right)$ and $\lambda_{2}=\lambda_{c}\left(i\right)$, respectively, where, as expected, $\lambda_{c}\left(1\right)\approx 0$. We will first describe, for a fixed p(X,Y), the relative variations in unique information as a function of $\lambda_{1}$ and $\lambda_{2}$. Then, we will compare the absolute values of unique information to the information globally processed by the system.

For all Figures from Figure 7 to Figure 9, the UI landscape partly mirrors the respective exact SR pattern of Figure 4, Figure 5 and Figure 6 (right). However, within the region where these latter figures answered a binary “no” to the question of exact SR, Figure 7, Figure 8 and Figure 9 reveal a sharply uneven variation in the violation of SR, where, for important ranges of trade-off parameters, the unique information is negligible comparative to others. For instance, even though Figure 5 (right) seems to indicate that SR does not hold for $\lambda_{1}<\lambda_{2}<\lambda_{c}\left(2\right)$, the corresponding UI in Figure 8 (right) is virtually zero on a large part of this set of parameters, while still peaking for $\lambda_{2}$ close to $\lambda_{c}\left(2\right)$. This richer structure of the unique information landscape is further evidenced by Figure 10, Figure 11 and Figure 12.

Moreover, the unique information landscapes seem shaped by the bottleneck trajectories. Most importantly, the influence of IB bifurcations on SR can be seen even more clearly with soft than with exact SR. In particular, in Figure 10, Figure 11 and Figure 12, it seems that along the lines where one of the trade-off parameters crosses a critical value, the UI often goes through discontinuities, or at least sharp variations in either $\lambda_{1}$, $\lambda_{2}$, or both directions. In particular, even though patterns widely vary across different example distributions p(X,Y), unique information can significantly *drop* when $\lambda_{2}$ crosses a critical value from below—a feature observed in both shown and non-shown examples. As we know that the unique information is continuous, the apparent discontinuity should be one of the bottleneck probability $q_{\lambda_{2}}\left(X,T_{2}\right)$ itself (see Proposition 7 and Remark 3). This is consistent with the observation from Section 2.3 that, at symbol splits, the trajectory of $q_{\lambda }\left(X|T\right)$ often seems to go through a discontinuity. Further, the fact that the sharp variation in UI is a *decrease* in this quantity (in increasing order of $\lambda_{2}$) is intuitively consistent with the fact that the bottleneck trajectory’s discontinuity often induces a sudden “widening” (in increasing order of λ) of
$$\begin{array}{c} H_{T}:=\mathrm{Hull}\left\{q\left(X|t\right),\;t\in T\right\}. \end{array}$$

Indeed, for fixed $\lambda_{1}$, when $\lambda_{2}$ crosses a critical value from below, the corresponding symbol split means that $H_{T_{2}}$ “widens” by opening up a new dimension, so it “more easily” encompasses $H_{T_{1}}$, yielding as a consequence a drop in unique information. Recalling our intuition (see Section 2.2) that $H_{T}$ describes the information content that a bottleneck *T* contains about the source *X*, the feature just described can be interpreted in the following way: the IB bifurcations seem to induce a sudden “expansion” (in increasing order of λ) of the information content carried by the bottleneck about the source, which makes the latter’s content more easily contain the information content of coarser bottlenecks.

Note, however, that these simple numerical results do not allow one to discriminate between the interpretation of the UI’s sharp variations at bifurcations as a discontinuity with regard to trade-off parameters, or a discontinuity of the UI’s *differential*. For instance, if the derivative with regard to $\lambda_{2}$ discontinuously takes a value close to −∞ for $\lambda_{2}$ slightly larger than some $\lambda_{c}$, then the UI graph can seem discontinuous at finite numerical resolution, even if, formally, only the UI’s differential is so. On the other hand, as an example, bifurcations can be characterised precisely as points of discontinuities of the derivatives, with regard to the trade-off parameter, of I(T;X) and I(T;Y) [43,75], even though the functions themselves are continuous [2,75]. A more involved analysis distinguishing discontinuities of UI from those of its differential is left to future work. In any case, the interpretation as a discontinuity of the differential rests on a weaker assumption, which is still sufficient for explaining the numerical results.

More generally, these results suggest that for a several-stage processing that is IB-optimal at each stage, to minimise the information discarded along stages, the trade-off parameters should rather lie close to well-chosen IB bifurcations. If this happens to be a general feature of the IB framework, it would have implications for incremental learning. Indeed, coming back to the modelling of embodied agents (see Section 1), for instance, it would mean that organisms that are poised close to information optimality by evolution should have a very specific structure of developmental learning, where the stages of learning should be discrete and determined by the right trade-off parameters.

Eventually, a last crucial feature was also satisfied on these minimal examples: whatever the structure of bottleneck trajectories, the maximal UI was significantly lower than the mutual information $I(X;T_{1},T_{2})$ between the external source *X* and the system’s internal representations $\left(T_{1},T_{2}\right)$. More precisely, for an extension $q(X,T_{1},T_{2})$ of $q_{\lambda_{1}}:=q_{\lambda_{1}}\left(X,T_{1}\right)$ and $q_{\lambda_{2}}:=q_{\lambda_{2}}\left(X,T_{2}\right)$ that achieves the minimum in (Equation 8), let us define
$$\begin{array}{c} \sigma \left(q_{\lambda_{1}},q_{\lambda_{2}}\right):=\frac{UI_{q_{\lambda_{1}},q_{\lambda_{2}}}\left(X:T_{1}\setminus T_{2}\right)}{I_{q}\left(X;T_{1},T_{2}\right)}. \end{array}$$ Note that decomposing $I_{q}\left(X;T_{1},T_{2}\right)$, where $q\in \Delta_{q_{1},q_{2}}$, with the chain rule for mutual information shows that this quantity only depends on $q_{\lambda_{1}}$ and $q_{\lambda_{2}}$: thus here, $\sigma (q_{\lambda_{1}},q_{\lambda_{2}})$ is indeed well-defined by $q_{\lambda_{1}}$ and $q_{\lambda_{2}}$. The maximum ratio over all trade-off parameters $\lambda_{1}<\lambda_{2}$ was typically of the order of 1% in our minimal experiments; for instance, it was 1.89%, 0.39%, 1.82%, 2.03%, 1.34%, and 0.31% for the IB problems corresponding to Figure 7, Figure 8, Figure 9, Figure 10, Figure 11 and Figure 12, respectively. Among all the (shown and non-shown) studied examples, it never exceeded 5.4%, and we did not notice an increase in this maximum ratio when the source or relevancy cardinalities were increased (the largest cardinalities that we experimented with were |X|=20, |Y|=10). In short, even though several-stage processing might incur a non-negligible loss of information optimality in the IB sense, these results suggest that this loss could often be significantly limited. Of course, here as in Section 2.3, on the one hand, the numerical results are purely phenomenological, and, on the other, it is at this stage far from being clear that the qualitative insights brought by these minimal experiments generalise well to more complex situations. However, they exhibit the potentially crucial qualitative features of exact and soft successive refinement in the IB framework, which can be targeted by further theoretical research.

## 4. Alternative Interpretations: Decision Problems and Deep Learning

The notion of successive refinement presented in this work builds on the intuition of the optimal incorporation of information. However, alternative interpretations can be given to the very same mathematical notion. First, thanks to the Sherman–Stein–Blackwell theorem [45,65], the rate-distortion-theoretic notion of SR can be shown to be equivalent to a specific order relation between the encoder of the finer bottleneck $q(T_{2}|X)$ and that of the coarser one $q(T_{1}|X)$, namely the *Blackwell order*. This point of view turns SR into an operational *decision-theoretic statement*; in short, there is SR when, for *any* task and *any* source distribution p(X), the optimal performance is better (or at least as good) when decisions are based on the output of $q(T_{2}|X)$ than when they are based on the output of $q(T_{1}|X)$. Second, the Markov chain (Equation 4) characterising successive refinement makes it directly relevant [46] to the IB analysis of deep neural networks [49,50,51,52,53,54,55,56]. In the next two sections, we make these connections explicit and relate them to this paper’s investigations.

### 4.1. Successive Refinement, Decision Problems, and Orders on Encoder Channels

Here, we show that exact and soft successive refinement can be, in the discrete case at least, understood in terms of optimally solving decision problems on arbitrary tasks, through orders on the encoder channels q(T|X) (or more precisely, pre-orders: i.e., we will consider binary relations that are reflexive and transitive). We will rely on [45], where these orders were considered.

Let us first make clear what we mean here by a decision problem. Consider a state variable *X* over a finite set X, another finite set A of possible actions, and a reward function u=u(x,a) that depends on both the value *x* of the state *X*, and the chosen action a∈A. The agent’s observation is not the state *X* itself, but only the output *T* of *X* through some stochastic channel κ:=p(T|X) (where we assume here that the observation space T is finite). To each observation-dependent policy π=π(A|T) corresponds an expected reward
$$\begin{array}{c} E_{\pi }\left(u\left(X,A\right)\right):=\sum_{t}p\left(t\right)E_{\left(X,A)∼p(X|t)\pi (A|t\right)}\left(u\left(X,A\right)\right), \end{array}$$
where p(X|t) is determined from κ:=p(T|X), p(X) through the Bayes rule, and p(X|t)π(A|t) denotes the product measure of p(X|t) and π(A|t). Solving the decision problem (p(X),A,u) for the observation channel κ means choosing a policy that yields an optimal expected reward
$$\begin{array}{c} R\left(p\left(X\right),\kappa ,u\right)\;:=\;\max_{\pi }\;E_{\pi }\left(u\left(X,A\right)\right). \end{array}$$

For instance, any Markov decision process can be seen as a decision problem as defined above (for discrete time and finite state-space, number of possible actions at each state, and horizon). In this case, X and T are the spaces of state trajectories and observation trajectories, respectively, that an agent can go through along one episode; A is the space of action sequences that can be chosen along the episode; and *u* is the cumulative reward, i.e., the (potentially discounted) sum of rewards obtained at each time-step in the episode. (See, e.g., [76] for more details on the terminology used in this example.)

We can now define the following order [45]:

**Definition** **10.**
*For two channels κ and μ, we write $\kappa {⊒}_{X}\mu$, if, for any decision problem (p(X),A,u), we have*

$$\begin{array}{c} R(p(X),\kappa ,u)\geq R(p(X),\mu ,u). \end{array}$$



In short, $\kappa {⊒}_{X}\mu$ means that, for any conceivable task based on any data distribution p(X) over the fixed data space X, the observation channel κ can yield a performance at least as good as that of the observation channel μ—if combined with a well-chosen policy. The second order is the *Blackwell order* [65]:

**Definition** **11.**
*For two channels κ and μ, we write $\kappa {⊒}_{X}^{\prime }\mu$ if there exists a channel η such that μ=η∘κ, where “∘” denotes the composition of channels, i.e., such that $M_{\mu }=M_{\eta }M_{\kappa }$, where $M_{\mu }$, $M_{\eta }$, and $M_{\kappa }$ are the column transition matrices corresponding to μ, η, and κ.*


It turns out that successive refinement can be characterised by either of these two orders, thanks to the Sherman–Stein–Blackwell theorem [45,65]. In other words, SR, which is *a priori* not a decision-theoretic statement, turns into one through its equivalence with the Blackwell order:

**Proposition** **8.**
*Let $0<\lambda_{1}<\lambda_{2}$. The following are equivalent:*

(i)
*There is successive refinement for parameters $\left(\lambda_{1},\lambda_{2}\right)$.*
(ii)
*There are bottlenecks $T_{1},T_{2}$ of respective parameters $\lambda_{1},\lambda_{2}$ such that*

$$\begin{array}{c} q\left(T_{2}|X\right){⊒}_{X}q\left(T_{1}|X\right). \end{array}$$

(iii)
*There are bottlenecks $T_{1},T_{2}$ of respective parameters $\lambda_{1},\lambda_{2}$ such that*

$$\begin{array}{c} q\left(T_{2}|X\right){⊒}_{X}^{\prime }q\left(T_{1}|X\right). \end{array}$$




**Proof.** Using the Markov chain characterisation (point (ii) in Proposition 1), the result is nothing more than a reformulation of Theorem 4 in [45] in the language of the present paper. Note that, to use this theorem, we need to assume that the source *X* is fully supported, but this is indeed an assumption that we are using along the whole paper because it does not incur any loss of generality (see Section 1.3). □

Let us highlight the intuitive meaning of Proposition 8. Point (ii) means that there is SR when the coarse representation $T_{1}$ can be retrieved by post-processing the finer representation $T_{2}$—which has implications in terms of feed-forward processing (see Section 4.2).

Now, the equivalence of SR with point (iii) relies on the mathematically deep part of the Sherman–Stein–Blackwell theorem [45], and provides a new operational meaning to SR. Namely, there is SR when, for *any* distribution p(X) on the source, and *any* reward function, the optimal performance is at least as good when the decisions are based on the output of $q(T_{2}|X)$, seen as an observation channel, than when they are based on the output of $q(T_{1}|X)$. Let us stress that the fact that $q(T_{2}|X)$ defines a finer bottleneck than $q(T_{1}|X)$ crucially depends on p(X,Y), i.e., on the specific source distribution p(X), and on how the latter relates to the specific relevancy variable through p(Y|X). Proposition 8 shows that SR describes a much more “universal” relation between the channels $q(T_{1}|X)$ and $q(T_{2}|X)$.

For example, assume that evolution poises the sensors of a given biological organism at optimality in the IB sense [10,16], i.e., if *X* is the environment, *S* some sensor’s output (e.g., a retina’s ganglion cells activation), and *Y* a behaviourally relevant feature (e.g., the edibility of food), then *S* is a bottleneck for p(X,Y). Successive refinement here means that if the sensor $S_{2}$ is finer than $S_{1}$ as a bottleneck for the fixed feature *Y* relevant to a particular task, then $S_{2}$ will afford to the organism—if combined with the right decision making—better performances than $S_{1}$ on any other task, for any other input distribution p(X). In other words, $S_{2}$ is then “universally better” than $S_{1}$, which is a very strong (and somewhat unexpected) generalisation.

Eventually, the unique information that we chose as our measure of soft SR has initially been thought precisely as measuring the deviation from the order “${⊒}_{X}$” (see arguments in [45]). Unique information can thus, for instance, be understood as quantifying the deviation from a finer IB-optimal sensor to be “universally better” than a coarser one.

### 4.2. Successive Refinement and Deep Learning

As suggested by Remark 1 and Proposition 8-(ii), successive refinement can be equally well understood in terms of feed-forward processing, an interpretation which is particularly relevant to deep neural networks. Indeed, while the information bottleneck theory of deep learning [49,50,51] is still under debate [52,53,54,55,56], our results can be connected to some of this theory’s specific claims concerning the benefits of hidden and output layers’ IB-optimality.

Let $L_{1},\cdots ,L_{n}$ denote the successive layers of a feed-forward deep neural network (DNN), which is fed with an input *X* and attempts to extract, within it, information about a target variable *Y*,thus satisfying the Markov chain [49]
(10)$$\begin{array}{c} Y-X-L_{1}-\cdots -L_{n}. \end{array}$$

One of the claims of the IB theory of DNNs [49,50,51] is that, once converged, a DNN’s hidden and output layers lie close to the information curve of the IB problem defined by p(X,Y), with each new layer corresponding to a coarser trade-off parameter. The performance and generalisation abilities of DNNs would rely on this IB-optimality of networks after training. While these claims have been challenged [52,77], the identified caveats have sparked a still ongoing line of research [54,55,56], which suggests that more nuanced versions of the initial claims might still hold. Most importantly for us here, numerical results suggest that layer-by-layer training with the IB Lagrangian as the loss function induces a performance on par with end-to-end training with cross-entropy loss [54], while recent theoretical work proved that the IB trade-off optimises a bound on the generalisation error [56]. In other words, the IB method seems to be relevant at least as a normative, if not descriptive, framework for DNNs. Thus, an interesting informationally optimal limit to compare a given DNN to is a sequence of variables $L_{1},\cdots ,L_{n}$ that(i)Satisfy the Markov chain (Equation 10); and(ii)Are each bottlenecks with source *X* and relevancy *Y*, for respective trade-off parameters $\lambda_{1}>\cdots >\lambda_{n}$.

However, it is not clear that variables satisfying those conditions even exist; actually, it is the case if and only if the IB problem is $\left(\lambda_{n},\cdots ,\lambda_{1}\right)$-successively refinable. Indeed, points (i) and (ii) are exactly the conditions of point (iii) in Proposition 1, with $T_{i}:=L_{n-i}$, and the order of trade-off parameters reversed as well. In this sense, the notion of exact successive refinement is relevant to deep learning; in particular—as suggested by the numerical results from Section 2.3—it might well be the case that there is successive refinement only for well-chosen combinations of trade-off parameters. In this case, an IB-optimal DNN should be designed and trained in such a way that its successive layers implement a compression corresponding to these well-chosen trade-off parameters.

**Remark** **4.**
*The single-letter formulation above mirrors, in large part, the asymptotic coding version of [46]. More precisely, Ref. [46] defines in asymptotic coding terms a feed-forward processing pipeline where each layer tries to extract, from the input coming from the previous layer, information about a potentially distinct relevancy $Y_{i}$. Theorem 2 in [46] shows that, for constant relevancy $Y_{i}:=Y$, the notion of “successive refinement” defined there by the authors happens to be equivalent to points (i) and (ii) above, and thus to our notion of “successive refinement”. In particular, the deep learning interpretation presented in this section also has an operational formulation in terms of asymptotic coding.*


Now, if exact SR describes the situation where each layer of a DNN can potentially reach the information curve, is our notion of soft SR also relevant to deep learning? Note that, here,

We know that the variables $L_{1},\cdots ,L_{n}$ must satisfy $X-L_{1}-\cdots -L_{n}$, i.e., we know that the joint distribution $q:=q(X,L_{n},\cdots ,L_{1})$ must be in $\Delta_{SR}$;And we want to know “how close” we can choose this joint distribution *q* to one whose marginals $q\left(X,L_{1}\right),\cdots ,q\left(X,L_{n}\right)$ coincide with bottleneck distributions $q_{1}:=q_{1}\left(X,T_{1}\right),\;\cdots ,q_{n}:=q_{n}\left(X,T_{n}\right)$, respectively, of parameters $\lambda_{1}>\cdots >\lambda_{n}$, respectively, i.e., we want to know how close we can choose *q* to the set $\Delta_{q_{1},\cdots ,q_{n}}$.

Thus, the quantity $D_{KL}(\Delta_{q_{1},\cdots ,q_{n}}\left|\right|\Delta_{SR,n})$ can also be interpreted as a measure of the deficiency of a DNN’s layers from all those simultaneously being bottlenecks. Note, however, that, in previous sections, we knew that any joint distribution $q(X,T_{1},\cdots ,T_{n})$ had to be in the extension set $\Delta_{q_{1},\cdots ,q_{n}}$, and wanted to know “how close” to the successive refinement set $\Delta_{SR,n}$, in the KL sense, we could choose it. On the contrary, in the case of DNNs, we know that any $q(X,T_{1},\cdots ,T_{n})$ must be in $\Delta_{SR,n}$—because the bottlenecks correspond to a DNN’s layer—and want to know “how close” to $\Delta_{q_{1},\cdots ,q_{n}}$ we can choose it.

From this perspective, the numerical results of Section 3.2 suggest interesting properties, or at least desirable features, of DNNs. First, if the fact that the UI is typically low generalises well from our minimal investigation to the much richer deep learning setting, this would imply that even in situations where a DNN’s successive layers cannot all lie exactly along the information curve, they might still be able to remain reasonably close to it. Second, the fact that UI (or its differential) seems to go through a discontinuity close to well-chosen bifurcations—such that the UI sharply drops when $\lambda_{2}$ crosses the bifurcation from below—suggests that, for each layer of the DNN to be individually as IB-optimal as possible, their corresponding trade-off parameters should each lie close to these IB bifurcations. This resonates with previous considerations suggesting that DNNs’ hidden layers should [49] or might indeed do [50] lie at IB bifurcations.

## 5. Limitations and Future Work

Our convex hull characterisation intertwines the question of exact SR with the more fundamental question of the structure of decoder curves
(11)$$\begin{array}{c} \left\{\left(\lambda \mapsto q_{\lambda }\left(X|t\right)\right),\;t\in T\right\} \end{array}$$
on the source simplex $\Delta_{X}$, a question for which the convexity approach to the IB problem [35,36,37,38,39] seems promising. In short, this approach reformulates the IB problem to that of finding the lower convex envelope of a well-chosen function $F_{\beta }$, defined on the source simplex $\Delta_{X}$, and parameterised by the information curve’s inverse slope β (see Section B.7). More precisely, bottlenecks are essentially characterised by the fact that the lower convex envelope must be achieved by convex combinations of the points $F_{\beta }\left(q\left(X|t\right)\right)$; this approach thus provides analytical tools for proving key properties of the set of trajectories (Equation 11), which would then have consequences for SR through the convex hull condition. Despite the limited scope of the result itself, the proof of Proposition 5 gives an example of such a fruitful interaction, thus suggesting a way forward for further theoretical progress. As a first step, one could try to use the convexity approach to the IB to prove our Conjecture 1 about the unicity, up to permutations and injectivity of q(X|T), for canonical bottlenecks and the strictly concave information curve. This would both simplify our convex hull characterisation of SR for the case of the strictly concave information curve (see Appendix D) and provide in itself a crucial property of the curves (Equation 11). Generally speaking, leveraging, through our convex hull characterisation, the convexity approach to the IB problem might allow one to (i) identify new wholly refinable IB problems, but also (ii) produce general methods to identify, for a given distribution p(X,Y), the combination of parameters for which exact SR holds.

It must be stressed that even though we motivate the successive refinement of the IB by diverse scientific questions in Section 2 and Section 4, in this work, we do not model any concrete system. Rather, our minimal numerical experiments target the qualitative exploration of the formalised problem. Our results might in turn be relevant for future modelling work (see the last paragraph of this section), but the most pressing aspect is to first develop further the theoretical and computational framework. In particular, it seems important to describe formally the apparent discontinuity of UI (or its differential) as a function of the trade-off parameters $\lambda_{1}$ and $\lambda_{2}$ at IB bifurcations (through that of the $q_{\lambda }\left(X,T\right)$ as functions of λ); to describe more formally why the UI tends to peak and then drop close to IB bifurcations; to provide global bounds on UI in general or as functions of the source and relevancy distribution p(X,Y); or to make formal the informal relationship between the “extent to which” the convex hull condition is broken, and variations in UI. Another interesting contribution would be to provide an asymptotic coding interpretation to unique information; indeed, the deviation from successive refinement is more classically quantified as a difference between asymptotic rates or distortions (see, e.g., [60]), and it is not clear whether or not this interpretation can be made for UI. Numerically speaking, one could design algorithms allowing for the computation of UI for continuous p(X,Y) and/or more than two processing stages. Indeed, the algorithm from [74] only encompasses the case of discrete variables and two processing stages. One could, for instance, take inspiration from [74] to formulate the quantity $D_{KL}(\Delta_{q_{1},\cdots ,q_{n}}\left|\right|\Delta_{SR,n})$ as a double minimisation problem over separate parameters, allowing for an alternating optimisation algorithm.

The deep learning interpretation of (exact and soft) SR depends crucially on some aspects of the ongoing debate on the IB theory of deep learning [49,50,51,52,54,55,56]. In this regard, it would be interesting to directly measure the unique information between different layers of a DNN or determine whether or not having the layers lying close to IB bifurcations does induce better performance or generalisation capabilities.

Let us point out that our framework considers that the source of information *X* and the target variable *Y* are the same along all processing stages. More general frameworks could allow for variations in either the source of information (as in the case in temporal series) or the target variable (as is the case in transfer learning). Frameworks for both these kinds of extensions have already been proposed [46,78], and it would be interesting to study if, in these cases as well, the specific nature of the IB problem imprints the informationally optimal limits of several-stage processing.

Eventually, we deem the interpretation in terms of the incorporation of information to be particularly relevant to modelling adaptive behaviour. For instance, for a given developmental or skill-learning problem on a given task, our framework could help in distinguishing situations where the choice of the successive representations’ complexity along incrementally learning the task does not matter (i.e., when there is successive refinement) from situations where these complexities must be minutely weighed, so as to avoid as much as possible the “waste” of cognitive work along the way (i.e., when the unique information is not negligible and unevenly distributed). In the latter case, our framework, once mature, might precisely describe those sequences of representations’ complexity that minimise the “waste” of cognitive work from one learning stage to another, thereby potentially identifying key stages of skill or developmental learning. Future work should keep in mind the horizon of identifying such qualitative features and producing measures capturing the relevant phenomena for experimental research in these areas.

## 6. Conclusions

Our approach in this paper is three-fold: to bring together in a common framework existing work on the exact successive refinement of the IB and related topics; to develop further this common framework, particularly through a geometric approach to the problem; and to then open up a line of research on the soft successive refinement of the IB.

The formal unity that we make explicit in this paper is mainly that between these three scientific questions: (i) that of informationally optimal incorporation of information—relevant in particular to developmental and skill learning; (ii) that of informationally optimal feed-forward processing—relevant in particular to describing and designing deep neural networks (DNNs); and (iii) that of channel order in statistical decision theory—which provides clear interpretations of distinct bottlenecks’ comparison in terms of universal informativeness of an agent’s sensor. Indeed, while we focused for most of the paper on the information incorporation interpretation, we saw in Section 4 that the two other ones are as legitimate as the first one.

Once the formal problem is motivated and set, we turn to the mathematical analysis of it. We first note that, for jointly Gaussian vectors (X,Y) or for deterministic p(Y|X), successive refinability can be easily drawn from existing IB literature [33,34]. Then, we propose a new geometric characterisation of SR, which builds on the intuition that what is “known” by a bottleneck is the convex hull of its decoder conditional probabilities. This new point of view, associated with an active approach that reformulates the IB problem as that of finding the lower convex envelope of a well-chosen function [35,36,37,38,39], provides a new tool for theoretical research on this topic. We exemplify this potential fertility by proving, thanks to the combination of our convex hull characterisation with the convexity approach to the IB, the successive refinability of binary source *X* and binary relevancy *Y* (Proposition 5). This convex hull characterisation also allows one to numerically investigate SR with a linear program, which can be helpful for computational studies on this topic. Our own minimal numerical experiments suggest that (i) successive refinement does not always hold for the IB, (ii) the successive refinement patterns are shaped by IB bifurcations, and (iii) even when successive refinement seems to break, sometimes it is “close” to being satisfied, in the sense of the convex hull condition being only “slightly” violated.

To formalise this latter intuition, we propose to soften the traditional notion of SR into a *quantification* of the loss of information optimality incurred by several-stage processing. For that purpose, we call on the measure of unique information (UI) used in [40]. Intuitively, this quantity measures the information that only the coarser bottleneck $T_{1}$, and not the finer one $T_{2}$, holds about the source *X*, and it can be generalised to an arbitrary number of processing stages. Our minimal experiments, in the case of two processing stages, unveil a rich structure of soft SR that was partially hidden by exact SR, which only makes the distinction between vanishing UI (if there is SR) and positive UI (if there is no SR). Even though the UI landscapes depend strongly on the distribution p(X,Y) that defines the IB problem, some qualitative features seem to emerge: (i) the “more” the convex hull condition is broken, the higher the unique information; (ii) the IB bifurcations crucially shape the UI landscape, with sharp decreases in unique information in particular when the finer trade-off parameter $\lambda_{2}$ crosses a bifurcation critical value; and (iii) in any case, this violation of successive refinement seems to always be mild compared to the system’s globally processed information.

The features exhibited by these numerical experiments offer a “first outlook” of potentially general properties of exact and soft successive refinement for the IB problem, thus providing a guide for future theoretical research. These potential properties might provide interesting perspectives on the scientific questions that motivate the formalism, particularly in terms of the incorporation of fresh information into already learned models, and deep learning. For instance, the apparently important role of bifurcations in exact and soft successive refinement suggests that informationally optimal several-stage learning or processing should ideally be organised along well-chosen “checkpoints” on the information plane. Moreover, if the loss of information optimality induced by this sequential processing is indeed typically low (even though not entirely negligible) for the IB framework, this could be taken as an indication that incremental learning might be made highly efficient. These potential features thus provide a strong incentive to bring the formal framework presented here closer to maturity—for instance, along the lines of research proposed in Section 5.

## Figures and Tables

**Figure 1 entropy-25-01355-f001:**
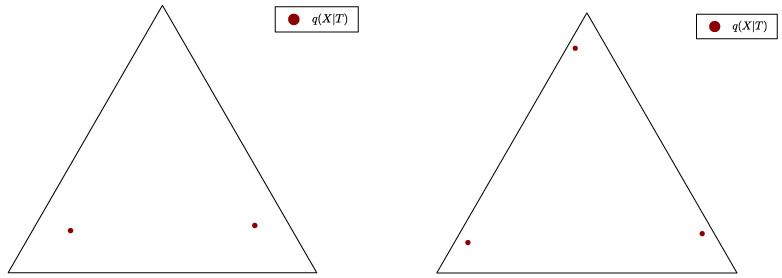
Examples of distributions q(X|T), visualised as families of points {q(X|t),t∈T} on the source simplex $\Delta_{X}$, where, here, |X|=3. Each of the triangle’s vertices represents the Dirac probability of some x∈X. The bottleneck’s effective cardinality is k=2 on the left and k=3 on the right.

**Figure 2 entropy-25-01355-f002:**
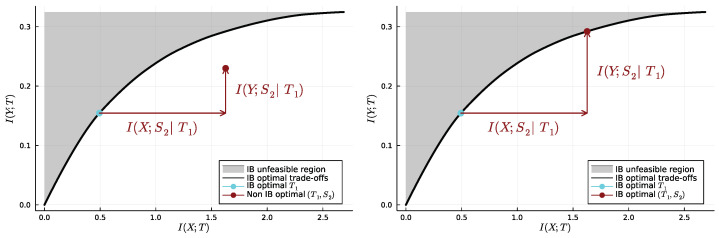
Successive refinement visualised on the information plane. On the left, adding the information from the variable $S_{2}$ (the supplement variable) is not efficient enough to achieve successive refinement. On the right, it is. See main text for details (the values of $I(X;S_{2}|T_{1})$ and $I(Y;S_{2}|T_{1})$ have been chosen arbitrarily to illustrate each case).

**Figure 3 entropy-25-01355-f003:**
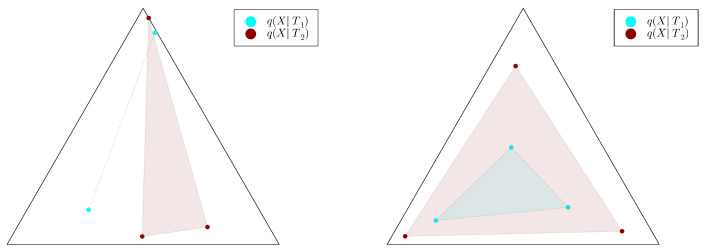
Illustration of the convex hull condition. The black triangle represents the source simplex $\Delta_{X}$ with, here, |X|=3, and the pointwise bottleneck decoder probabilities {q(X|t),t∈T} are represented on it (in cyan for the coarser bottleneck $T_{1}$ and in red for the finer one $T_{2}$). The convex hull of the respective families of points are shaded with the corresponding color. On the left, the condition is not satisfied; on the right, it is.

**Figure 4 entropy-25-01355-f004:**
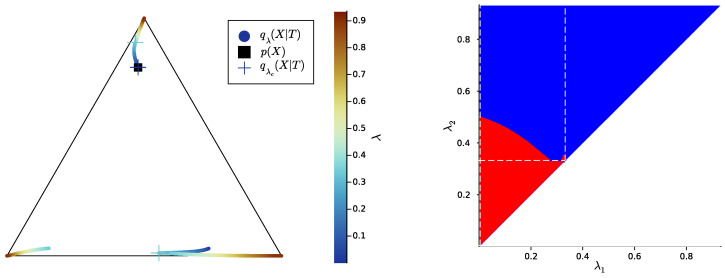
Left: bottleneck trajectories for an example distribution p(X,Y) such that |X|=|Y|=3, i.e., trajectory of $q_{\lambda }\left(X|T\right)$, represented as the family of points $\left\{q_{\lambda }\left(X|t\right),\;t\in T\right\}$ on the source simplex $\Delta_{X}$, as a function of λ=I(X;T) (crosses: value of $q_{\lambda_{c}}\left(X|T\right)$ just before a symbol split at a critical parameter $\lambda_{c}$, where the crosses’ color corresponds to the value of $\lambda_{c}$). The conditional distribution $q_{\lambda }\left(X|T\right)$ is defined by the single point p(X) when λ=0 (dark blue cross on the black square), or by two distinct points between the first and second symbol splits (dark blue to cyan), or by three distinct points after the second symbol split (cyan to red). Note the discontinuity of $q_{\lambda }\left(X|T\right)$ at each symbol split (without the discontinuity, the trajectory around a symbol split would look like a branching). Right: corresponding SR pattern, i.e., corresponding output for the convex hull condition (blue: satisfied; red: not satisfied; dashed white lines: critical values $\lambda_{c}\left(i\right)$ of either $\lambda_{1}$ or $\lambda_{2}$). For instance, the critical value $\lambda_{c}\left(2\right)\approx 0.33$ corresponds, on the bottleneck trajectories (left), to the symbol split from two to three symbols (cyan crosses). Note that $\lambda_{c}\left(1\right)\approx 0$. The respective p(Y|X) corresponding to this figure and to Figure 5 and Figure 6 are plotted in Appendix E.

**Figure 5 entropy-25-01355-f005:**
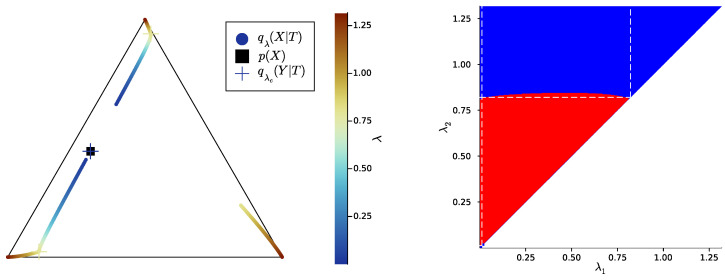
Same as Figure 4, with a different example distribution p(X,Y) such that |X|=|Y|=3.

**Figure 6 entropy-25-01355-f006:**
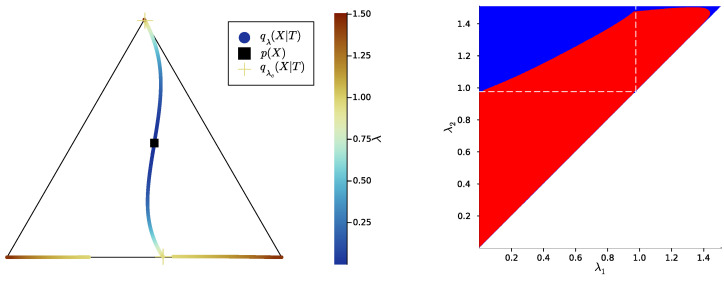
Same as Figure 4, with a different example distribution p(X,Y) such that |X|=|Y|=3.

**Figure 7 entropy-25-01355-f007:**
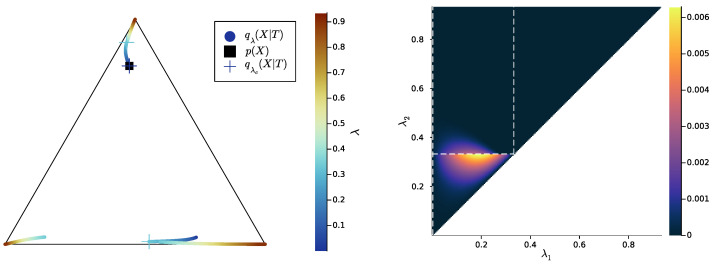
Left: example trajectory of $q_{\lambda }\left(X|T\right)$ as a function of λ=I(X;T) (crosses: value of $q_{\lambda_{c}}\left(X|T\right)$ just before a symbol split at a critical parameter $\lambda_{c}$). Right: corresponding unique information, in bits (color), expressed as a function of the pair of trade-off parameters (white dashed lines indicate critical values $\lambda_{c}\left(i\right)$ of either $\lambda_{1}$ or $\lambda_{2}$.). For instance, the critical value $\lambda_{c}\left(2\right)\approx 0.33$ (right) corresponds, on the bottleneck trajectories (left), to the symbol split from two to three symbols (cyan crosses). The respective p(Y|X) corresponding to this figure and to Figure 8 and Figure 9 are plotted in Appendix E.

**Figure 8 entropy-25-01355-f008:**
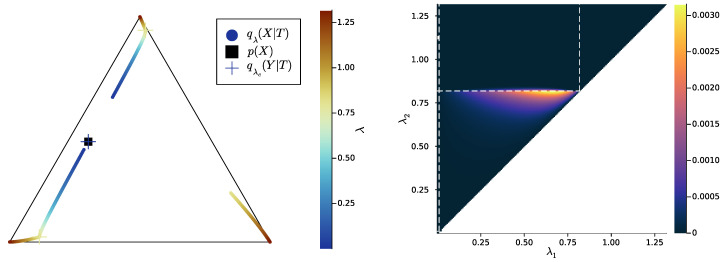
Same as Figure 7, where the example distribution p(X,Y) is that of Figure 5.

**Figure 9 entropy-25-01355-f009:**
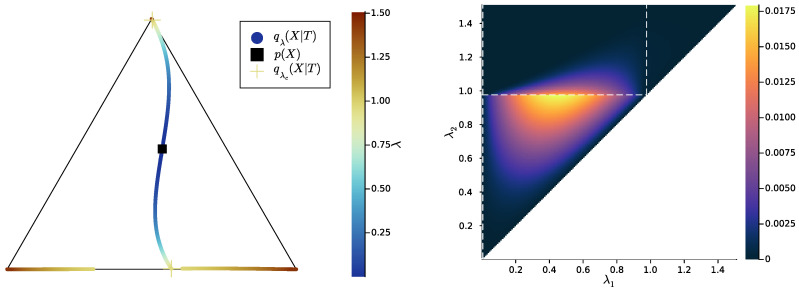
Same as Figure 7, where the example distribution p(X,Y) is that of Figure 6.

**Figure 10 entropy-25-01355-f010:**
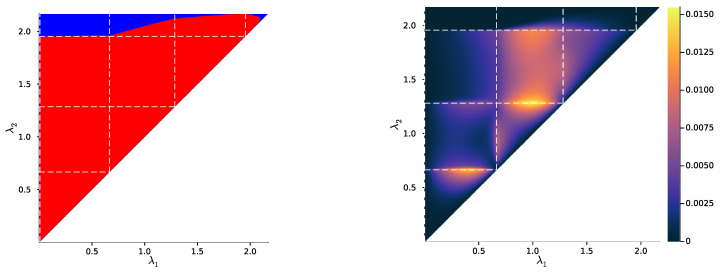
New example of an exact SR pattern and the corresponding UI landscape over trade-off parameters $\lambda_{1}<\lambda_{2}$, where, here, |X|=5 and |Y|=3. Left: exact SR pattern, i.e., output for the convex hull condition (blue: satisfied, red: not satisfied). Right: corresponding UI landscape, in bits (color). White dashed lines indicate critical values $\lambda_{c}\left(i\right)$ of either $\lambda_{1}$ or $\lambda_{2}$. Note that (i) the binary notion of exact SR (left) filters out most of the structure unveiled by UI (right), (ii) the UI landscape seems highly impacted by IB bifurcations, and (iii) the UI is in any case always small, even though not entirely negligible. See main text for more details.

**Figure 11 entropy-25-01355-f011:**
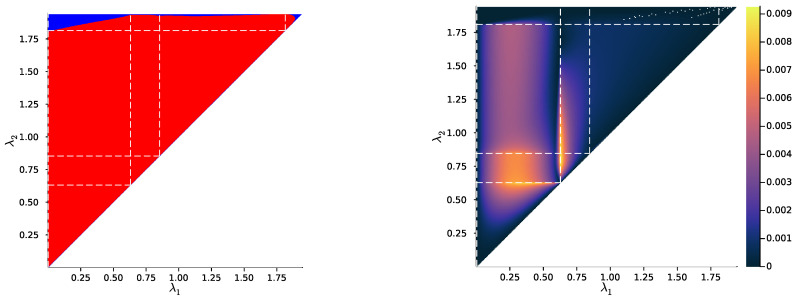
Same as Figure 10, with a new example distribution p(X,Y), where, here, |X|=5 and |Y|=3. Besides the white orthogonal dashed lines, other white dots correspond to values of $\left(\lambda_{1},\lambda_{2}\right)$ for which the algorithm did not converge (see main text for a comment on this lack of convergence).

**Figure 12 entropy-25-01355-f012:**
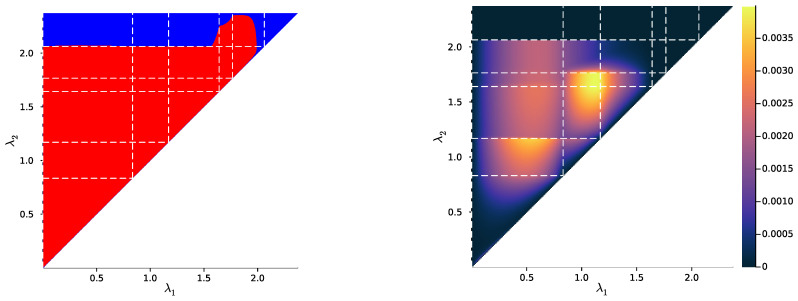
Same as Figure 10, with a new example distribution p(X,Y), where, here, |X|=7 and |Y|=5.

## Data Availability

The code that we used for this work can be found at https://gitlab.com/uh-adapsys/successive-refinement-ib/, along with the explicit values of the example distributions p(X,Y) that we used to generate our figures.

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
