# Peer review of "Exact and Soft Successive Refinement of the Information Bottleneck"

_entropy, 2023, doi:10.3390/e25091355_

Round 1

Reviewer 1 Report

The authors considered the successive refinement version of the information bottleneck (IB) problem. A detailed study of the properties of the IB tradeoff is given and several specific cases are investigated in detail. A softer notion of successive refinability is then introduced which quantifies the loss if the setting is not completely successive refinable.

The manuscript is well written and presented in general.

1. My main concern is the relation with existing work. The problem considered in this submission is the single-letter version of the IB problem, while several existing works considered the operating meaning of the same problem (i.e., multi-letter physical systems), and had similar general characterizations at the end. For example, the authors already mentioned that Proposition 1 is "mostly an equivalent reformulation of the known results" in [29,30]. In fact, I believe the exact same results were given in those references, and also in the following reference:

Tuncel, Ertem. "Capacity/storage tradeoff in high-dimensional identification systems." IEEE Transactions on Information Theory 55.5 (2009): 2097-2106.

I would recommend the authors make clear what results were already known and what is new here. I would believe the formulation itself and Proposition 1 are not really novel, but Propositions 2-4 are new. 

2. Another concern is the relationship between IB and deep learning. Several observations and hypotheses in [47,48] had been mostly disapproved, see for example: 

Goldfeld, Ziv, and Yury Polyanskiy. "The information bottleneck problem and its applications in machine learning." IEEE Journal on Selected Areas in Information Theory 1.1 (2020): 19-38.

I would suggest revising the part related to this aspect accordingly.

3. The issue of loss of successive refinability had been studied in the RD setting; see this article

Lastras, Luis, and Toby Berger. "All sources are nearly successively refinable." IEEE Transactions on Information Theory 47.3 (2001): 918-926.

A short discussion on the relation is needed.

Reviewer 2 Report

The authors apply the idea of soft and exact successive refinement (SR) to the IB, notice relationships to bifurcations, provide some “minimal” numerical examples, and make ties to “deep learning”.   This work is theoretical.   I do have suggestions to: better explain how this work is different than previous work; help with the presentation;  provide the reader with some small, easy to understand examples to help understand definitions, concepts and results.

1.      I appreciate that SR is defined in abstract and early on at line 86.   Unfortunately, it is not clear what is meant by exact vs soft SR until later in the paper.   Lines 324-5 appear to tie exact SR with Definition 5, but then “soft” SR is not defined until lines 613-620.  Please define early in the paper what is meant by exact vs. soft SR.  

2.      Lines 123-4, here, early in the paper, please provide more info re: “it often goes through sharp variations at the bifurcations [3942] undergone by the bottlenecks over λ.”  What is meant by “sharp variations”?

3.      Line 124 – “loss of information” at bifurcation, this depends on whether one is annealing forward or reverse, can you make this clear?  

4.      Lines 177-180, I appreciate that you compare your work to [44].  But please provide more details describing how [44] is different than you.  Isn’t [44] SR more general than the SR you present?   In other words, is your work simply a special case of [44]?   Is there an advantage to what you do vs [44]?

5.      Line 185: provide more details here in the paper what is meant by “trained deep neural networks should lie close to IB bifurcations” – you do flush this out a little more later in section 4.

6.      Lines 194 and 375-377 refer to a recent 2022 publication [31] that makes the link between “renormalization group theory” and bifurcations, although few details are provided.   Author’s reference [69] from 2012 used the maximal subgroup structure of the symmetric group Sm (where m is the order of the alphabet for T) to identify where bifurcating solutions for IB occur (Theorem 5.3) and to identify the bifurcating directions (Corollary 3.4) of the IB solutions when using the Lagrangian formulation.   How does [31] characterize bifurcations?   Are the bifurcation characterizations found in [31] and [69] similar?   Are the approaches similar?

7.      Lines 225-6, what does “Unless explicitly stated otherwise, the variables in this paper are assumed either all discrete or all continuous.” mean?  I think you mean that all of X, Y and T are all one or the other.  Please re-word.

8.      Lines 227-9, is it really necessary to state “We denote by the same symbol, e.g. H(X), both the discrete entropy and the differential entropy; and similarly use the same symbol, e.g. I(X;Y), for both the discrete and continuous mutual information.”   In my opinion, this statement can be removed.

9.      Line 232 and again 239-240, 244, I do not understand “T is a bottleneck, i.e., a solution to (1)” and “T is a solution to the primal IB problem (1)”  In fact, a conditional probability q(|) is the optimal solution to (1), as the authors state on line 418 “the IB problem (1) from an optimisation over the encoder channels q(T|X)”, please clarify.

10.  Lines 260-3 where definitions of effective cardinality and canonical form are provided, afterwards please give an example for a small problem, for example when |X|=3 and |T|=2 or 3 or some other convenient example.   These definitions are confusing because Def 1 defines effective cardinality with respect to the distinct values of q(X|T=t) which suggests that k <= |X|, but then Def 2 suggests that k=|T|.

11.  Lines 273-4, consider rewriting the statement “the marginals of qi and qj over their common coordinates” in the definition, it is confusing, although the example after the definition made clear what the authors are trying to say.

12.  Line 312 where SR is defined, afterwards please give an example for a small problem, for example when |X|=3 and |T|=2 or 3 or some other convenient example.   The example should show rvs T_1 and S_2

13.  What is the point of Proposition 3 that describes SR for the very simplistic scenario when Y is a deterministic function of X?    How does it contribute to the rest of the paper?

14.  Line 362 “Proposition 1 is mostly an equivalent reformulation of already known results.”   Please make clear what’s new in Prop 1.

15.  Lines 460-464 after Proposition 4 were important to help me understand the injectivity assumption, thanks.

16.  Line 551, when describing Figure 3, “For λ 0, the qλ(X|t) all coincide with the source distribution p(X)”, so then why don’t all the trajectories in Fig 1 start at a black square?   Right now, only the q(X|t) in the top corner starts at a black square.   This question pertains to all Figures that show trajectories.

17.  Line 560, “each q(X|t)”, I think the authors mean q(X|t) for all $t\in {\cal T}$ with |T|=3, please clarify.

18.  I am confused by line 573 where λc(i) is defined with i discussed in Notation 2, please consider adding more details or an example, λc(1) and  λc(2) seem to be the only references in the text.

19.  Figure 3 (and other Figures) show a cross “just before a symbol split” for 2 of the 3 trajectories (top and right), why not for the trajectory in the bottom left?  

20.  Should Figures 3, 5, 6, 8 legends use the notation q(|) as opposed to p(|)?

21.  Definition 6, why not provide the formula for $D_{KL}(r_1||r_2)$?

22.  Lines 671-672, even though there is no way to exactly compute $D_{KL}$ with respect to those spaces for n>2, surely there are ways to estimate it, perhaps using the approach described by [72]?

23.  Lines 766-771, when describing the decision problem with respect to actions, rewards and policies, please follow with an example.

24.  Line 785, please define what is meant by “composition of channels”.

25.  Lines 806-809, “Then successive refinement between T1 and T2 means … that, whatever the environment’s state and whatever the task, the finer sensor can always yield better performance than the coarser one, if it is leveraged with the right policy.  This is not an obvious conclusion …”.  This statement is hard to understand without a concrete example, please provide a small example.    And why is it not an obvious conclusion?

26.  Lines 859-863, please reword, not sure what is being swapped, what holds and what lacks.

27.  Line 869, reword “the lack of SR often sharply dropping close to bifurcations”

28.  Lines 897 – 900, I agree that this is a major limitation of the authors work “in this work we do not model any concrete system. Rather, our minimal numerical experiments target the qualitative exploration of the formalised problem. Our results might in return be relevant for future modeling work”.   Can the authors point out some specific systems where they think their approach would be helpful?

29.  Lines 975-6, not sure what is meant by “was partially hidden by the either true or false notion of exact SR”, please rewrite.

There are some minor grammatical issues identified in my comments.

Round 2

Reviewer 1 Report

My concerns have been adequately addressed.

Reviewer 2 Report

The authors have satisfactorily addressed my comments.